



# Glyoxal tropospheric column retrievals from TROPOMI, multi-satellite intercomparison and ground-based validation

Christophe Lerot[1], François Hendrick[1], Michel Van Roozendael[1], Leonardo M.A. Alvarado[2,3], Andreas Richter[3], Isabelle De Smedt[1], Nicolas Theys[1], Jonas Vlietinck[1], Huan Yu[1], Jeroen Van Gent[1], Trissevgeni Stavrakou[1], Jean-François Müller[1], Pieter Valks[4], Diego Loyola[4], Hitoshi Irie[5], Vinod Kumar[6], Thomas Wagner[6], Stefan F. Schreier[7], Vinayak Sinha[8], Ting Wang[9], Pucai Wang[9], Christian Retscher[10]

[1] Royal Belgian Institute for Space Aeronomy (BIRA-IASB), Brussels, Belgium
[2] Alfred Wegner Institute, Helmholtz Center for Polar and Marine Research, Bremerhaven, Germany
[3] University of Bremen, Institute of Environmental Physics, Bremen, Germany
[4] Institut für Methodik der Fernerkundung (IMF), Deutsches Zentrum für Luft und Raumfahrt (DLR), Oberpfaffenhofen, Germany
[5] Center for Environmental Remote Sensing, Chiba University, Japan
[6] Max Planck Institute for Chemistry (MPIC), Mainz, Germany
[7] Institute of Meteorology and Climatology, University of Natural Resources and Life Sciences, Vienna, Austria
[8] Indian Institute of Science Education and Research Mohali, Mohali, India
[9] Institute of Atmospheric Physics, Chinese Academy of Sciences (CAS), Beijing, China
[10] European Space Agency, ESRIN, Frascati, Italy

*Correspondence to:* Christophe Lerot, Christophe.Lerot@aeronomie.be

**Abstract.** We present the first global glyoxal (CHOCHO) tropospheric column product derived from the TROPOspheric Monitoring Instrument (TROPOMI) on board of the Sentinel-5 Precursor satellite. Atmospheric glyoxal results from the oxidation of other non-methane volatile organic compounds (NMVOCs) and from direct emissions caused by combustion processes. Therefore, this product is a useful indicator of VOC emissions. It is generated with an improved version of the BIRA-IASB scientific retrieval algorithm relying on the Differential Optical Absorption Spectroscopy (DOAS) approach. Among the algorithmic updates, the DOAS fit now includes corrections to mitigate the impact of spectral misfits caused by scene brightness inhomogeneity and strong $NO_2$ absorption. The product comes along with a full error characterization, which allows providing random and systematic error estimates for every observation. Systematic errors are typically in the range of $1-3 \times 10^{14}$ molec/cm² (~30-70% in emission regimes). Random errors are larger (>$6 \times 10^{14}$ molec/cm²) but can be reduced by averaging observations in space and/or time. Benefiting from a high signal-to-noise ratio and a large number of small-size observations, TROPOMI provides glyoxal tropospheric column fields with an unprecedented level of details.

Using the same retrieval algorithmic baseline, glyoxal column data sets are also generated from the Ozone Monitoring Instrument (OMI) on Aura and from the Global Ozone Monitoring Experiment-2 (GOME-2) on board of Metop-A and Metop-B. Those four data sets are intercompared over large-scale regions worldwide and show a high level of consistency. The satellite glyoxal columns are also compared to glyoxal columns retrieved from ground-based Multi-Axis (MAX-) DOAS instruments at nine stations in Asia and Europe. In general, the satellite and MAX-DOAS instruments provide consistent glyoxal columns both in terms of absolute values and variability. Correlation coefficients between TROPOMI and MAX-DOAS glyoxal columns range between 0.61 and 0.87.





The correlation is only poorer at one mid-latitude station, where satellite data appears low biased during
wintertime. The mean absolute glyoxal columns from satellite and MAX-DOAS generally agree well for
low/moderate columns with differences less than $1\times10^{14}$ molec/cm². A larger bias is identified at two sites where
the MAX-DOAS columns are very large. Despite this systematic bias, the consistency of the satellite and MAX-
DOAS glyoxal seasonal variability is excellent.

## 1. Introduction

Exposure to poor air quality kills millions of people annually (e.g. Vohra et al., 2021; World Health Organization,
2016) due to natural and human emissions of a large range of particulate matters and gases, including among
others nitrous oxides (NOx) , sulphur dioxide, carbon monoxide, methane and volatile organic compounds
(VOCs). The latter, in combination with NOx, play a significant role in the secondary production of tropospheric
ozone (Jacob, 2000), which is highly toxic for the respiratory system and also contributes to global warming
because of its absorption in the thermal infrared. Global measurements of atmospheric concentrations of the ozone
precursors is therefore crucial. The number of VOCs that can be found in the atmosphere is manifold, but only a
few of them can be probed using remote sensing techniques. For example, formaldehyde (HCHO) measurements
have been used in many studies as a proxy for probing emissions of non-methane VOCs of biogenic, pyrogenic
and anthropogenic origin (e.g. Abbot et al., 2003; Barkley et al., 2013; Bauwens et al., 2016; Beekmann and
Vautard, 2010; Curci et al., 2010; Jin et al., 2020; Marais et al., 2012; Palmer et al., 2006; Stavrakou et al., 2016;
Wells et al., 2020).
With a lifetime of a few hours, glyoxal (CHOCHO) is another short-lived VOC that can be detected remotely,
offering the potential to provide information on Non-Methane VOC (NMVOC) emissions. Over the past few
years, an increasing number of studies (e.g. Cao et al., 2018; Chan Miller et al., 2017; Fu et al., 2008; Li et al.,
2016; Liu et al., 2012; Stavrakou et al., 2009, 2016; Wittrock et al., 2006)have exploited glyoxal measurements
from space, often in combination with formaldehyde. Being produced from similar sources, those two species are
complementary as they have different production yields. For example, the oxidation of aromatics produces glyoxal
with a much higher yield than formaldehyde (Cao et al., 2018). Although being both mostly produced via the
oxidation of other VOCs, direct emissions from anthropogenic and fire activities also occur, and contribute more
to the glyoxal global budget than to the formaldehyde one (Stavrakou et al., 2009b, 2009a). This motivated many
studies to investigate the ratio of glyoxal to formaldehyde concentrations or columns as a possible metric to
discriminate between different types of VOC emissions (e.g. Chan Miller et al., 2014; DiGangi et al., 2012; Hoque
et al., 2018; Kaiser et al., 2015; Vrekoussis et al., 2010). Glyoxal measurements are also essential for establishing
the global budget of secondary organic aerosols (SOAs). Indeed, with a high solubility in water, glyoxal undergoes
heterogeneous uptake on aerosols and cloud droplets where the subsequent aqueous-phase chemistry forms SOA
(Chan et al., 2010; Fu et al., 2008; Hallquist et al., 2009; Knote et al., 2014; Li et al., 2016; Volkamer et al., 2007).
Glyoxal has three absorption bands in the visible spectral range that have been exploited to remotely retrieve
information on its atmospheric abundance using the Differential Optical Absorption Spectroscopy method
(DOAS, Platt and Stutz, 2008) applied to ground-based (e.g. Benavent et al., 2019; Hoque et al., 2018; Javed et
al., 2019; Schreier et al., 2020), air-borne (e.g. Kluge et al., 2020; Volkamer et al., 2015), ship-borne (e.g. Behrens
et al., 2019; Sinreich et al., 2010) and space-based instruments. The first global glyoxal tropospheric column





observations from space have been realized by Wittrock et al. (2006) using nadir measurements from the
SCIAMACHY (SCanning Imaging Absorption spectroMeter for Atmospheric CartograpHY) instrument. Based
on this pioneering work, different glyoxal data products were derived from the Global Ozone Monitoring
Experiment-2 (GOME-2) (Lerot et al., 2010; Vrekoussis et al., 2009) and from the Ozone Monitoring Instrument
(OMI) (Alvarado et al., 2014; Chan Miller et al., 2014). All those different products rely on a similar DOAS
approach, but generally differ from each other by the choice of the fit settings and of the auxiliary input data.
In general, the glyoxal optical depth is very low (< 5e-4), typically one order of magnitude smaller than the $NO_2$
optical depth in the same spectral range. This results in retrievals prone-to-noise, requiring to average many of
them to extract meaningful glyoxal signals. With an enhanced spatial resolution resulting in a number of
observations more than ten times larger than provided by its predecessor OMI, the TROPOspheric Monitoring
Instrument (TROPOMI), operating since 2017, allows observing weak atmospheric absorbers with an
unprecedented level of spatio-temporal details. This has been illustrated by Alvarado et al. (2020a) who
investigated the large amounts of formaldehyde and glyoxal emitted by the intense North-American wildfires in
August 2018 as observed by TROPOMI for several days and over long distances. Theys et al. (2020) have
evaluated the respective contributions to the hydroxyl radical production in fresh fire plumes from nitrous acid,
VOCs and other sources with the support of different TROPOMI data sets, including the glyoxal data product
described here.
This work presents the latest version of the BIRA-IASB scientific glyoxal tropospheric column retrieval algorithm
that has been applied to three years of TROPOMI measurements, and also to data from the predecessor nadir
instruments OMI and GOME-2A/B. The quality of the TROPOMI glyoxal retrievals is investigated with (1) a
global intercomparison of the satellite glyoxal data products generated with a common algorithm and (2)
comparisons with independent glyoxal measurements from a series of Multi-AXis DOAS (MAX-DOAS)
instruments located at nine stations in Asia and Europe.
After a brief introduction of the satellite instruments used in this study in Section 2, the retrieval algorithm and its
different steps are described in Section 3, with emphasis on the updated and innovative aspects compared to
heritage studies. This section also presents the typical random and systematic errors associated to the retrievals
and how they are estimated for each individual measurement. Section 4 presents the evaluation of the inter-satellite
consistency by comparing both seasonal global spatial patterns as seen from different instruments as well as
monthly mean time series and seasonal cycles in a series of selected large-scale regions. Finally, Section 5 presents
validation results based on MAX-DOAS data.
**2. TROPOMI and other nadir-viewing satellite sensors**
TROPOMI was launched on 13 October 2017 on board of the Sentinel-5 precursor platform. It flies on a sun-
synchronous Low Earth Orbit (LEO) with an ascending node crossing the equator at the local time of 13:30. In
the series of Sentinel missions from the European Union Copernicus programme, it is the first one dedicated to
atmospheric composition. The instrument operates in a nadir viewing mode and measures Earthshine radiances
and solar irradiances in the ultraviolet (UV), visible, near infrared and short infrared spectral bands. It aims at
providing column amounts of a number of key pollutants, such as ozone ($O_3$), $NO_2$, $SO_2$, HCHO, CO, $CH_4$ as well
as cloud and aerosol parameters. TROPOMI offers a quasi-daily global coverage at the unprecedented spatial



resolution of 3.5x5.5 km² (3.5x7 km² before August 2019) in the UV-visible spectral range. It is an imager-type
instrument using a two-dimensional Charge Coupled Device (CCD) for the light measurements, the detector
columns being used for the spectral resolution while the rows are binned to resolve spatially the 2600 km across-
track swath into 450 individual ground pixels. The spectral resolution of the instrument is about 0.5 nm and offers
a remarkably high signal-to-noise ratio of about 1500 in band 4 (405-500 nm) used in this study. More details on
the instrument and its performance can be found in (Kleipool et al., 2018; Ludewig et al., 2020; Schenkeveld et
al., 2017; Veefkind et al., 2012). The TROPOMI measurements allow to derive the vertical columns of multiple
species, some of them not included among the operational products listed above. Glyoxal is one of them and the
details on how its column quantities are retrieved will be described in the next section.
The TROPOMI design strongly inherits from past nadir-viewing sensors, and in particular from the Ozone
Monitoring Instrument (OMI) that we use to evaluate the TROPOMI glyoxal product presented in this work. OMI
(Levelt et al., 2006) is also an imager instrument and flies on an early afternoon orbit since October 2004. The
OMI swath, divided into 60 across-track pixels with a size varying from 13x24 km² (at nadir) to 13x150km² (at
the edges), allowed a daily global coverage before being limited in 2008 by the so-called row anomaly. The latter
consists in a modification of the signal recorded by OMI at specific rows, due to a mechanical obstruction of the
field of view, and leads to lower quality spectral measurements (Torres et al., 2018). We also exploit spectral
measurements from the Global Ozone Monitoring Experiment-2 (GOME-2) instruments aboard the Metop-A and
Metop-B platforms. In contrast to OMI and TROPOMI, the GOME-2 instruments (Munro et al., 2016) fly on
early morning LEOs with local equator crossing times around 09:30 and are scanning spectrometers, meaning that
across-track pixels are successively sounded. The scan is divided into 24 pixels for a total swath of 1920 km,
providing global coverage in 1.5 day. Each pixel has a size of 80x40km². After the launch of Metop-B, the GOME-
2A swath was reduced to 960 km in July 2013, leading to ground pixel two times smaller.

## 3. Description of the Algorithm

The algorithm for retrieving tropospheric vertical columns of glyoxal relies on a classical DOAS approach (Platt
and Stutz, 2008). This approach consists first in fitting measured optical depths in an optimized spectral window
to derive the so-called slant column densities *SCDs* (atmospheric concentration integrated along the effective light
path) of the absorbers. The latter are thereafter converted into vertical column densities *VCDs* (concentration
vertically integrated from the satellite ground pixel up to the top of the atmosphere) with air mass factors obtained
by modelling the radiative transfer through the atmosphere. An additional background correction procedure is
often applied for weak absorbers such as glyoxal in order to reduce as much as possible the presence of systematic
biases caused by spectral interferences.
The glyoxal algorithm presented here largely inherits from past developments for predecessor nadir-viewing
satellite sensors (Alvarado et al., 2014, 2020; Chan Miller et al., 2014; Lerot et al., 2010; Vrekoussis et al., 2009;
Wittrock et al., 2006). In the following subsections, we further describe each algorithmic component, with
emphasis on its specificities. The retrievals are provided with estimates for the random and systematic errors,
which are discussed in subsection 3.4.



### 3.1. DOAS fit

To exploit the glyoxal absorption bands, we use a fitting window from 435 to 460 nm encompassing the two most intense bands, which has shown in the past to provide reliable results (Barkley et al., 2017; Lerot et al., 2010). This has been confirmed by sensitivity tests carried out by Alvarado et al. (2014) and Chan Miller et al. (2014). Owing to its low optical depth ($<5x10^{-4}$), any poorly fitted feature in the radiance measurements may affect the retrieved glyoxal SCD. It is therefore crucial to account for any physical or instrumental effect in order to optimise the fit quality as much as possible. Different aspects of the algorithm contribute to achieve this.

The wavelength grids of the measured spectra are recalibrated before the actual DOAS fits with a cross-correlation procedure (Danckaert et al., 2017; De Smedt et al., 2018) during which the position of the lines in the measured irradiance spectrum is fitted to an external solar atlas (Chance and Kurucz, 2010), convolved to the satellite spectral resolution. This recalibration procedure is done once per orbit and separately for every detector row of the instrument.

Although the DOAS fit generally uses an irradiance as the reference spectrum, it is common practice, in the case of weak tropospheric absorbers, to replace it by a mean radiance spectrum recorded in a remote region where the concentration of the gas of interest is low (e.g. De Smedt et al., 2018). This allows reducing the presence of systematic biases caused by spectral interferences and/or instrumental limitations. In particular, the use of one separate mean radiance spectrum per detector row minimizes the presence of so-called stripes in the product typical of imager-type instruments such as OMI or TROPOMI. Here we compute those mean radiance spectra on a daily basis by averaging for each row all spectra located within the equatorial Pacific Ocean (15°S-15°N; 120°W-180°W).

The selected settings for the DOAS fits rely on the aforementioned past studies and are summarized in Table 1. The latest available cross-sections for species absorbing in the selected fitting window are included in the fit, i.e. $O_3$, $NO_2$, $O_2$-$O_2$, water vapour and liquid water in addition to glyoxal. Note that the water vapour cross-section is based on the HITRAN2012 database (Rothman et al., 2013) as we found that the latest HITRAN2016 version (Gordon et al., 2017) led to poorer fit quality. The temperature dependence of the $NO_2$ absorption is taken into account by including a second cross-section, taken as the difference between $NO_2$ cross-sections reported at 2 temperatures (220 and 294K) as proposed by Alvarado et al. (2014) and Chan Miller et al. (2014) for their respective OMI glyoxal products. Consistently with Alvarado et al. (2014), we found that fitting the liquid water optical depth in the glyoxal fitting window performs as well as fixing it to a value previously determined in a larger spectral interval as proposed in the past (Lerot et al., 2010). A number of additional cross-sections are included in the fit to consider (1) Inelastic scattering (Ring effect) introduces high-frequency structures that are treated as a pseudo-absorber (Chance and Spurr, 1997); (2) Intensity offsets in the spectra, caused for example by residual straylight, are corrected for by fitting the inverse of the reference spectrum (Danckaert et al., 2017); (3) heterogeneity of the scene brightness may also introduce high frequency structures, which are considered with pseudo-cross-sections (more details hereafter). All those cross-sections are generated at the instrumental spectral resolution by using the key data Instrumental Spectral Response Functions provided for all individual detector rows. During the DOAS procedure, the earthshine radiance spectrum is further aligned with the reference, by allowing it to be shifted and stretched in wavelength. In addition, the DOAS fit procedure includes a spike removal



scheme as described in Richter et al. (2011) enabling to filter out from the fit individual corrupted radiance
measurements, and hence to reduce the noise in the product.
**Table 1 : Absorption cross-sections and settings used for the retrieval of glyoxal slant columns**

| Fitting interval | 435-460 nm |
|---|---|
| **Absorption cross-sections** | |
|    Glyoxal | Volkamer et al. (2005) |
|    Ozone | Serdyuchenko et al. (2014), 223K |
|    $NO_2$ | Vandaele et al. (1998), 220K and 294K, $I_0$ effect-corrected (Aliwell et al., 2002) |
|    $O_4$ ($O_2$-$O_2$) | Thalman and Volkamer (2013), 293K |
|    $H_2O$ (vapour) | Rothman et al. (2013), 293K |
|    $H_2O$ (liquid) | Mason et al. (2016) |
|    Scene Heterogeneity | 2 pseudo-absorbers (Richter, 2018) – Internally generated |
|    Ring effect | Pseudo-absorber (Chance and Spurr, 1997; Wagner et al., 2009) |
| **Other parameters** | |
|    Polynomial | 3rd order |
|    Intensity offset correction | 1st-order offset (additional cross-section taken as the inverse of the reference spectrum) |
|    Earthshine wavelength shift | 1st-order shift |
| **Reference spectrum ($E_0$)** | Daily average of radiances, per detector row, selected in equatorial Pacific (Lat: [-15° 15°], Long: [180°-240°]) |


**3.1.1. Scene heterogeneity**
Any intensity variation within the probed scene taking place perpendicularly to the instrumental slit (i.e. along
track) leads to perturbations of the instrumental spectral slit function (ISRF) (Noël et al., 2012; Voors et al., 2006).
Richter et al. (2018)have shown that those perturbations lead to a degradation of the $NO_2$ DOAS spectral fit quality
and to systematic biases on the retrieved slant columns. Such abrupt intensity changes occur for example along
the coasts, mountains or cloud edges. Glyoxal retrievals are also affected by such scene heterogeneity as illustrated
in Figure 1 over the Horn of Africa and Middle East. This figure shows in the panel (a) that the root mean square
(RMS) of the DOAS fit residuals is systematically higher along the coasts but also over land where contamination
by broken clouds or abrupt elevation changes cause discontinuities in brightness fields. The stripes visible in this
figure are due to the smaller pixel size (end hence lower signal-to-noise ratio) on the edges of the across-track
field of view. The panel (c) shows that there are some collocated artificial patterns (positive/negative biases) in
the mean retrieved glyoxal slant column field. The latter result from spectral interferences with the signature
introduced by the ISRF distortion. Richter et al. (2018) showed that those spectral interferences can be
significantly reduced with additional cross-sections in the DOAS fit scaling the possible scene heterogeneity
signature. Those cross-sections are generated with a statistical analysis of the fit residuals for many observations


in a remote region as a function of the level of scene heterogeneity. The latter can be computed using radiance
measurements at higher spatial resolution available in the TROPOMI level-1 data at a limited number of
wavelengths. Following this approach, two additional cross-sections have been added to the DOAS baseline and
both the fit residuals and the identified glyoxal biases have been reduced as illustrated in the right panels (b) and
(d) of Figure 1. This effect is particularly visible along coasts and mountains but also over lands where some
pseudo-noise caused by persistent broken clouds is also largely reduced. Note that a third cross-section derived
from the mean residuals of homogeneous scenes is also added, which explains why the fit RMS are also reduced
(but less drastically) in homogeneous scenes. This cross-section has no impact on the retrieved glyoxal SCDs and
allowed mostly isolating systematic residuals due to scene heterogeneity only for the pseudo cross-sections
creation.

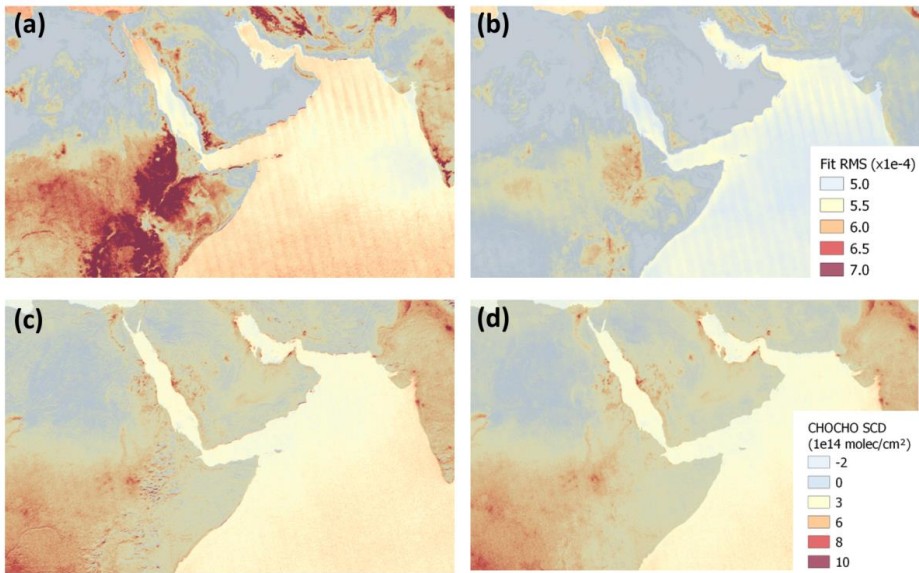


**Figure 1 : Impact of scene brightness heterogeneity on glyoxal retrievals in the fitting window 435-460 nm over the**
**Horn of Africa and Middle-East. The panels (a) and (b) show mean fit residuals RMS for the year 2019 without and**
**with (left and right) pseudo-cross sections to correct for spectral signatures introduced by scene heterogeneity. The**
**panels (c) and (d) show the corresponding mean glyoxal slant column densities. Only observations with cloud**
**fractions less than 20% are considered.**
**3.1.2. Empirical correction for strong $NO_2$ absorption**
The DOAS approach assumes that the wavelength dependence of the effective light path within the fit interval
can be neglected. Although this assumption is generally reasonable, it may fail in case of strong absorption by one
(or more) species, of which the slant column density becomes dependent on the wavelength (Puķīte et al., 2010).
In that case, fitting the optical depth of that species by a simple scaling of its cross-section is inaccurate and the
fit quality is degraded. Puķīte et al. (2010) have shown that fitting additional cross-sections resulting from a Taylor





expansion of the wavelength-dependent slant column corrects for its variability within the fit window. As
mentioned before, the high sensitivity of glyoxal retrievals to potential sources of misfit was a motivation to
further investigate its sensitivity to extreme $NO_2$ concentration levels.
For this purpose, synthetic spectra were generated at a spectral resolution of 0.5 nm with the radiative transfer
model SCIATRAN (Rozanov et al., 2005) for a satellite nadir-viewing geometry and two different solar zenith
angles. In those simulations, inelastic scattering was neglected and a large range of tropospheric $NO_2$ columns
was covered by scaling the $NO_2$ a priori profile. The TROPOMI DOAS baseline described above was then applied
to those simulated spectra in order to retrieve CHOCHO SCDs and evaluate the error as a function of the $NO_2$
SCD as illustrated in Figure 2. Results clearly point to a CHOCHO SCD error increasing with the $NO_2$ SCD. Note
that the exact error magnitude may change slightly depending on the $NO_2$ vertical distribution and on the actual
atmospheric content. On the other hand, adding the so-called Pukite cross-sections (Puķīte et al., 2010) to account
for the wavelength-dependence of the $NO_2$ SCD significantly reduces the errors.

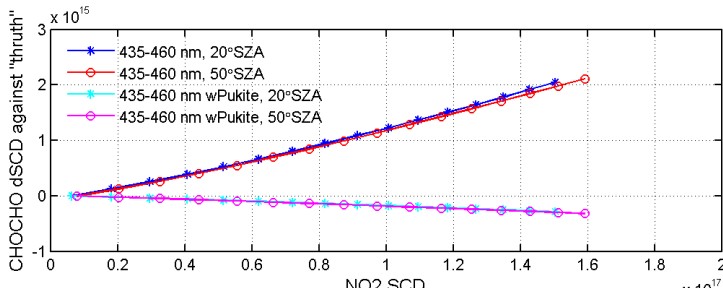


**Figure 2 : Absolute error (in molec/cm²) on the retrieved CHOCHO SCD as a function of the $NO_2$ SCD for simulated**
**spectra in a nadir-viewing satellite geometry and for two solar zenith angles. The reference "true" CHOCHO SCD is**
**taken as the value retrieved for the lowest NO2 SCD scenario. The error increases with the $NO_2$ SCD when Pukite**
**cross-sections are not included in the fit, but remains small otherwise.**
On this basis, the impact of adding the Pukite cross-sections to the DOAS baseline has been investigated using
one month of TROPOMI data. A wintertime period was chosen (December 2019) to favour the number of
observations with large $NO_2$ concentrations, in particular in China but also in other megacities in the Northern
Hemisphere. Figure 3 (upper panel) displays the monthly mean $NO_2$ SCDs in December 2019, and (middle panel)
the mean impact on the retrieved CHOCHO SCDs of introducing the Pukite terms in the DOAS spectral fit
baseline. The CHOCHO SCD differences caused by the Pukite terms are also plotted as a function of the $NO_2$
SCDs to better visualize the correlation (lower panel). For regions with enhanced $NO_2$ concentrations ($>2 \times 10^{16}$
molec/cm²) (e.g. China, India, Teheran), the Pukite cross-sections lead to a systematic reduction of the CHOCHO
SCDs, consistent with the closed-loop tests described above. A small improvement of the fit quality is found (not
shown). Unexpectedly, the impact of those additional cross-sections on the CHOCHO SCDs can also be non-
negligible in regions with low $NO_2$ columns: positive differences are for example observed over equatorial oceans,
but also over South America and Africa. The correlation plot of Figure 3 clearly shows these two regimes. While
the impact of the Pukite cross-sections on the glyoxal retrievals is understood and reliable for large $NO_2$ SCDs,





their influence at low $NO_2$ SCD is more questionable and likely results from spectral interferences occurring
between the different fitted spectra (e.g. with the Ring signature), which introduces additional noise in the product.
To avoid this, rather than fitting additional cross-sections, we introduce an empirical correction applied to the
glyoxal SCDs. This correction consists in subtracting from the glyoxal SCD an $NO_2$-SCD dependent value,
directly prescribed from the linear regression fit through the sensitivity test results for all observations worldwide
from December 2019, with $NO_2$ SCDs larger than $2x10^{16}$ molec/cm² as illustrated in Figure 3 (c). It is worth
noting that the regression fit results agree well with the glyoxal SCD errors estimated from the simulations
presented above (Figure 2). For extreme pollution conditions such as what can be found in China during
Wintertime, this correction may lead to glyoxal column reduction up to 30%.



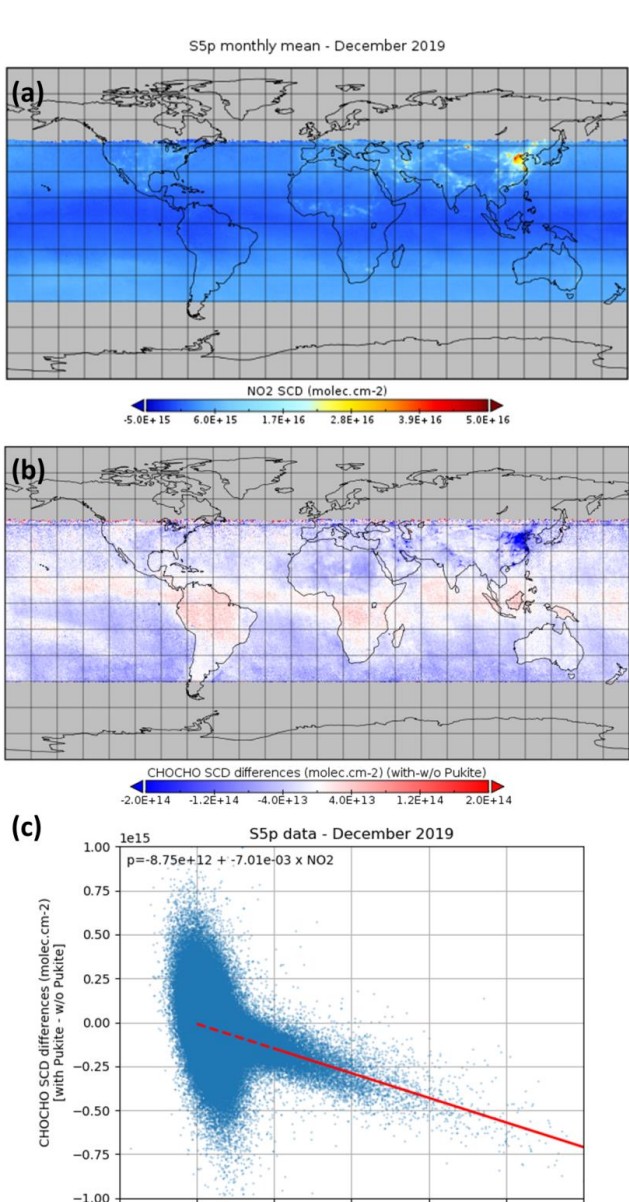


**Figure 3 : (a) Monthly mean NO₂ SCDs retrieved from TROPOMI data in December 2019. Panel (b) illustrates the CHOCHO SCD absolute differences (molec/cm²) due to the incorporation of the Pukite et al. (2010) cross-sections in the DOAS spectral fit and panel (c) shows the correlation between those differences and the NO₂ SCDs. The red line corresponds to a linear regression fit through all points with NO₂ SCD larger than $2 \times 10^{16}$ molec/cm².**





### 3.2. Air Mass Factor computation

The computation of the air mass factor (AMF) used to convert the retrieved glyoxal slant columns (SCD) to
vertical columns (VCD) relies on the formulation of Palmer et al. (2001), which decouples the radiative transfer
through the atmosphere from the vertical distribution of the gas of interest. Radiative transfer simulations are
performed with the vector model VLIDORT at the middle of the fitting window (448 nm) to compute so-called
altitude-dependent air mass factors or box-AMFs representing the sensitivity of the slant column to a small
concentration change at any altitude. The AMF is obtained as the weighted mean of those box-AMFs using as
weights the vertical distribution of the glyoxal concentration.
Typically, the sensitivity of nadir-viewing UV-Visible instruments is reduced in the lowermost atmospheric layers
because of Rayleigh scattering. However, this sensitivity depends strongly on the observation geometry, on the
surface reflectivity and altitude and on the presence of clouds. For example, the sensitivity is generally further
reduced for low sun elevation. For this reason, retrievals with solar zenith angles larger than 70° are filtered out.
We use a pre-computed five-dimensional look-up table of Box-AMFs spanning all observation conditions (see
Table 2) and from which appropriate values are linearly interpolated for every TROPOMI observation. This
interpolation uses as input the observation angles provided in the level-1 data, surface elevation taken from the
GMTED2010 topography (Danielson and Gesch, 2011) and surface albedo extracted from the OMI minimum
Lambertian Equivalent Reflectivity climatology (Kleipool et al., 2008). The spatial resolution of the latter
database (0.5°x0.5°) is coarse compared to the TROPOMI footprint and neglects anisotropy, which may introduce
significant errors (Lorente et al., 2018). However, at the time of writing, it is the only database available at the
S5p overpass time although new Lambertian Equivalent Reflectivity climatologies relying on past works (e.g.
Loyola et al., 2020; Tilstra et al., 2021, 2017) are currently being prepared. On the other hand, the level of noise
in glyoxal retrievals generally requires averaging in space and/or time which in turn will reduce part of those error
sources. We also neglect the impact of clouds and aerosols on the radiative transfer. Instead we apply a stringent
cloud filtering approach: only observations with an effective cloud fraction (as retrieved in the same spectral range
and provided in the TROPOMI operational $NO_2$ product (van Geffen et al., 2019) lower than 20% are conserved.
This approach is motivated by the fact that glyoxal slant columns tend to be biased high over bright scenes because
of poorly understood residual spectral interferences (e.g. with the Ring signature). Similarly, scenes covered by
snow and ice are also discarded.

**Table 2 : Granularity of the Box-AMF look-up table**

| Parameter name | Grid of values |
|---|---|
| Solar zenith angle [deg] | 0, 10, 20, 30, 40, 45, 50, 55, 60, 65, 70, 72, 74, 76, 78, 80, 85 |
| Line of sight zenith angle [deg] | 0, 10, 20, 30, 40, 50, 60, 65, 70, 75 |





| Relative azimuth angle [deg] | 0, 45, 90, 135, 180 |
|---|---|
| Surface albedo | 0, 0.01, 0.025, 0.05, 0.075, 0.1, 0.15, 0.2, 0.25, 0.3 0.4, 0.6, 0.8, 1.0 |
| Surface pressure [hPa] | 1063.10, 1037.90, 1013.30, 989.28, 965.83, 920.58, 876.98, 834.99, 795.01, 701.21, 616.60, 540.48, 411.05, 308.00, 226.99, 165.79, 121.11 |


The a priori glyoxal vertical profile shapes necessary to perform the AMF computations are provided by the global
Chemistry Transport Model MAGRITTE developed at BIRA-IASB, which inherits from the IMAGES model
(Bauwens et al., 2016; Müller and Brasseur, 1995; Stavrakou et al., 2009b, 2013). This model runs at 1°x1°
resolution and calculates the distribution of 182 chemical compounds, of which 141 species undergo transport.
The modelled troposphere is vertically divided in 40 levels between the surface and the lower stratosphere and
meteorological fields are provided by the ECMWF ERA-5 analyses. The chemical mechanism and deposition
scheme have been recently updated (Müller et al., 2018, 2019). Anthropogenic NMVOC emissions of are
provided by the EDGAR 4.3.2 inventory (Huang et al., 2017) for the year 2012. Biomass burning emissions are
obtained from the Global Fire Emission Database version 4 (GFED4s) (Van Der Werf et al., 2017). The emissions
of isoprene and monoterpenes are calculated using the MEGAN-MOHYCAN model (Guenther et al., 2012;
Müller et al., 2008). The model also incorporates biogenic emissions of methanol, methyl-butenol, ethylene,
ethanol, acetaldehyde, formaldehyde and acetone, as well as oceanic emissions of methanol, acetone, acetaldehyde
and alkyl nitrates (Müller et al., 2019). The global source of glyoxal in the model amounts to 47 Tg/yr (in 2013),
of which about 4 Tg/yr are due to direct biomass burning emissions, and 18, 6, 9 and 9 Tg/yr are due to the
atmospheric degradation of isoprene, acetylene, aromatics and monoterpenes, respectively (Müller et al., 2019).
To account for the difference in spatial resolution between the model and the observations, a priori profiles are
rescaled to the effective satellite pixel surface elevation using the formulation proposed by Zhou et al. (2009).
Enhanced glyoxal concentrations have been detected over oceans in several studies (Coburn et al., 2014; Lerot et
al., 2010; Sinreich et al., 2010), but current models cannot reproduce this. For this reason, over oceans, we use an
a priori glyoxal concentration profile measured with an air-borne MAX-DOAS instrument over the Pacific Ocean
during the TORERO campaign (Volkamer et al., 2015).
**3.3. Background correction**
As already mentioned, systematic (row-dependent) biases in the retrieved SCDs often remain due to small residual
interferences with spectral signatures from other absorbers or due to instrumental effects. In the particular case of
pushbroom imaging instruments such as OMI/TROPOMI, across-track row-dependent biases (so-called stripes)
often occur due to the imperfect calibration of the different CCD detector rows. To reduce those biases, a
background correction using observations in a remote reference sector is generally applied as part of the retrieval
algorithm (e.g. Alvarado et al., 2014; Chan Miller et al., 2014; Lerot et al., 2010; Richter and Burrows, 2002; De
Smedt et al., 2018). The principle of this background correction is to add offset values to the retrieved SCDs to





ensure that the resulting mean VCD in a clean remote region match an a priori known tropospheric glyoxal
column. Here we use the Pacific Ocean as reference sector with a constant reference VCD of $1 \times 10^{14}$ molec/cm$^2$.
This value was chosen according to independent measurements performed in this region (Sinreich et al., 2010)
since current global models fail to reproduce remote sensing glyoxal levels observed over oceans (Fu et al., 2008;
Myriokefalitakis et al., 2008; Stavrakou et al., 2009b).
The background correction is applied on a daily basis in different steps:

1.  First, a destriping procedure such as proposed in Boersma et al. (2007) is applied consisting in an offset
    correction determined separately for each instrumental row, and relying on clear sky observations from
    the Equatorial Pacific Ocean (15°S-15°N, 165°E-220°E). The offset corrections are added to all glyoxal
    SCDs worldwide, considering their respective row.

2.  Additionally to the high frequency stripes, a broadband row-dependent structure, of which the shape also
    depends on the latitude, was identified as illustrated in Figure 4, panel (a). This figure compares the row-
    dependence of mean uncorrected VCDs in the Pacific Ocean at Equatorial and Northern mid-latitudes.
    The two curves are somehow anti-correlated, meaning that the destriping correction based on equatorial
    latitudes only as applied in step 1 is not sufficient and even reinforces the mid-latitude structure. The
    second step of the background correction aims thus at reducing this broadband row-dependent structure
    at all latitudes while maintaining the mean latitudinal distribution of the measured background glyoxal
    columns.  For this, Pacific Ocean measurements (40°S-40°N, 165°E-220°E) are binned per 20° in latitude
    and in groups of 15 rows in a 2-dimensional matrix. For this step, we use reference VCDs depending on
    the latitude and resulting from the averaging of the binned VCDs along the row dimension. A
    corresponding 2-dimensional matrix of SCD offset corrections is then computed in order, once applied
    to the binned VCDs, the corrected values match the reference VCDs. Interpolation through this correction
    matrix provides offsets to be applied to all SCDs retrieved worldwide.

3.  Finally, the overall level of the product is adjusted with a single offset correction to ensure that the mean
    of all clear-sky VCDs within the full reference sector (40°S-40°N, 165°E-220°E) is equal to $1 \times 10^{14}$
    molec/cm$^2$. Panel (b) of Figure 4 shows how the identified row dependence in the VCDs at different
    latitudes has been reduced. The general level of the columns has also been adjusted.

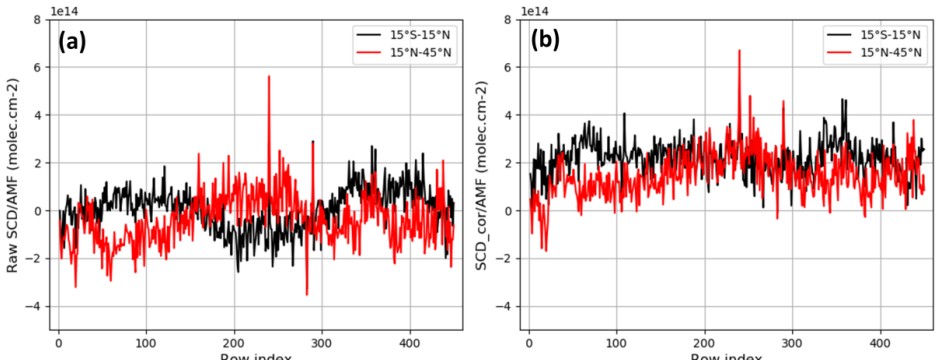

**Figure 4 : Row-dependence of the glyoxal vertical columns of S5p orbit #5877 (December, 1st 2018) averaged in an equatorial latitude band and in a Northern mid-latitude band. (a) No background correction is applied; (b) a latitude-dependent background correction is applied.**

### 3.4. Uncertainty estimates

Glyoxal tropospheric column retrievals are affected by many sources of uncertainties in the different components of the algorithm. The low glyoxal optical depth makes its retrieval highly sensitive to measurement noise and to spectral interferences with strong absorption signatures of other species or with instrumental features. Although the measurement noise can be reduced by averaging column retrievals from individual observations, spectral interferences generally lead to residual systematic errors (biases), which cannot be easily eliminated. The background correction described above aims at reducing those biases, but it has its own limitations. For example, the reference glyoxal tropospheric column within the reference sector is poorly known. In addition to spectral fit errors, there are also significant errors associated to the air mass factor calculations, mostly originating from input parameters uncertainties. For estimating the total glyoxal column error, we assume that the different error components are uncorrelated and can be summed quadratically as in (Boersma et al., 2011; Lerot et al., 2010; De Smedt et al., 2008, 2018). If the glyoxal vertical column $N_v$ is expressed as

$$N_v = \frac{N_s - \overline{(N_{s,0} - N_{v,0,ref} \times M_0)}}{M} \tag{1}$$

with $N_s$ the retrieved slant column, $M$ the AMF, $\overline{(N_{s,0} - N_{v,0,ref} \times M_0)}$ the background correction term where $N_{s,0}$, $M_0$, $N_{v,0,ref}$ are the slant columns, AMF, and the reference vertical column within the reference sector, the total glyoxal vertical column error can be written as

$$\sigma^2_{N_V} = \frac{1}{M^2} \left( \sigma_{N_s}^2 + N_v^2 \sigma_M^2 + \sigma_{N_{s,0}}^2 + N_{v,0,ref}^2 \sigma_{M,0}^2 + M_0^2 \sigma^2_{N_{v,0,ref}} \right) \tag{2}$$

where $\sigma_{N_s}$, $\sigma_M$ and $\sigma_{N_{v,0,ref}}$ are the errors on the slant column, the air mass factor and the reference value used in the background correction, respectively. In the following subsections, we discuss the different contributions to each of those terms. Errors can affect the retrievals randomly or systematically (biases). While the main random error is caused by the propagation of the instrumental photon detector shot noise on the measured radiances, the other error components are considered as being systematic.





### 3.4.1. Slant column uncertainties

As mentioned above, the radiance measurement noise directly propagates into the glyoxal slant column retrieval and leads to large random errors $\sigma_{N_{s,rand}}$ (or precision) due to the low glyoxal optical depth. Those are easily estimated using the fit residuals RMS and the covariance matrix of the cross-sections included in the fit (Danckaert et al., 2017). In the visible spectral range, the TROPOMI signal-to-noise ratio is about 1600 over dark scenes. This leads to a glyoxal VCD precision (i.e. $\sigma_{N_{s,rand}}$/AMF) in the range of $6\text{-}10 \times 10^{14}$ molec/cm² as illustrated in Figure 5, panel (d). This range of values is consistent with the scatter observed in the retrieved glyoxal SCDs in regions without any significant glyoxal source. Over bright scenes, for example covered by clouds or snow, those errors significantly drop because of the increased signal-to-noise ratio. For individual observations, random errors dominate and averaging is needed to extract meaningful glyoxal signals.

There are also systematic errors associated to the DOAS spectral fit that are mainly dominated by absorption cross-section uncertainties, by interferences with other species ($O_4$, liquid water, Ring …), or by other effects such as residual stray light. Those contributions are difficult to assess and can only be estimated from sensitivity tests (Lerot et al., 2010). In general, this error term can be as high as $2\text{–}3 \times 10^{14}$ molec/cm². However, the use of a radiance as reference in the DOAS fit and the application of a background correction removes a large part of the systematic error in the slant column fit (see section 3). As those corrections are not always sufficient to eliminate completely the SCD systematic errors due to local conditions (local pollution, residual clouds,…), we set $\sigma_{N_{s,syst}}$ to $1 \times 10^{14}$ molec/cm².

### 3.4.2. AMF uncertainties

The errors on the air mass factor depend on the input parameter uncertainties and on the sensitivity of the air mass factor to each of them. This contribution can be broken down into the squared sum (Boersma et al., 2011; Lerot et al., 2010; De Smedt et al., 2018) as

$$\sigma_{M,syst}^2 = \left(\frac{\partial M}{\partial A_s} \cdot \sigma_{A_s}\right)^2 + \left(\frac{\partial M}{\partial s} \cdot \sigma_s\right)^2 + (0.15M)^2 \tag{3}$$

where $\sigma_{A_s}$ and $\sigma_S$ are typical uncertainties on the surface albedo and profile shape, respectively.

The contribution of each parameter to the total air mass factor error depends on the observation conditions. Therefore, a small table of air mass factor derivatives spanning all observation conditions was computed using VLIDORT, considering glyoxal box profile shapes with different effective heights.

The AMF error component related to the surface reflectivity (1$^{st}$ term of Eq. ((3)) is calculated using an estimated uncertainty on the albedo $\sigma_{A_s}$ of 0.02 (Kleipool et al., 2008). Note that this uncertainty can be occasionally larger, in particular at high latitudes where snow falls may cause abrupt changes in scene albedo. The uncertainty associated to the a priori profile shapes (the smoothing error) used in the retrieval is more difficult to assess, especially due to the scarcity of independent glyoxal profile measurements. For every observation, an effective height corresponding to the a priori glyoxal profile used in the AMF calculation is derived and used to extract the appropriate AMF derivative and $\sigma_S$ is taken equal to 50hPa.



Formulation ($3$) is valid for clear sky pixels and the stringent cloud filtering we use. However, residual clouds
undoubtedly impact the radiative transfer and generally shield the lowermost atmospheric layers. Therefore, we
anticipate that the clear sky assumption generally leads to a low bias on the retrieved glyoxal columns in case of
residual clouds. On the other hand, the spectral interferences over bright (cloudy) scenes as discussed in section
3.2 impact the retrievals the other way round. The third term in equation ($3$) accounts for possible errors in the
AMF model itself, including the neglect of aerosols and clouds, wavelength dependence,…, and is estimated to
be 15% of the air mass factor (Lorente et al., 2017).
**3.4.3. Background correction uncertainties**
Although the background correction is designed to overcome systematic features/deficiencies of the slant column
fitting, some errors are also associated to this procedure. In particular, systematic errors on the reference slant
columns and their air mass factors are propagated into the computed correction values. Also, there is an uncertainty
related to the reference glyoxal vertical column value in the reference sector. The three last terms of Eq. ($2$)
represent the total background correction uncertainty in which $\sigma_{N_{s,0}}$ is the systematic slant column error fixed to
$1 \times 10^{14}$ molec/cm² (see above section 6.5.1), and $M_0$ and $\sigma_{M_0}$ are the air mass factors and their associated errors
within the reference sector. In practice, those quantities are treated similarly as the reference slant columns (i.e.
binned in latitude and row bins – see section 3.3). $\sigma_{N_{v,0,ref}}$ represents the error associated to the reference value
$N_{v,0,ref}$ and is fixed to $5 \times 10^{13}$ molec/cm².
**3.4.4. Total systematic uncertainties**
Figure 5, panel (c) shows the estimated mean VCD systematic errors for the month of June 2018 when all
systematic error sources are combined together using Eq. ($2$). Note that the conversion of the AMF error into an
absolute vertical column error (2[nd] term of the equation) requires this error to be multiplied by the corresponding
vertical column. Because of the high level of noise in the product, using the retrieved column for this would lead
to a strong overestimation of the systematic error. To circumvent this, we use instead pre-computed climatological
glyoxal noise-free VCDs.
Total glyoxal VCD systematic errors are generally in the range $1$-$3 \times 10^{14}$ molec/cm², corresponding to about 30-
70% for emission regimes (columns larger than $2 \times 10^{14}$ molec/cm²). Note that pixels strongly contaminated by
clouds (cloud fraction > 20%) or covered by snow/ice are discarded. Systematic errors are expected to be large
for those pixels mainly due to spectral interference effects (see section 3.2) and also because the information
content on glyoxal is reduced in case of cloud shielding. Figure 5, panel (b) shows monthly mean AMFs for the
same month. Small AMFs are generally caused by a priori profiles peaking near the surface, which makes the
retrieval more sensitive to albedo uncertainties and to a lesser extent to the a priori profile shape uncertainties.
This explains the anti-correlation between the AMFs and the systematic errors. In contrast, large AMFs are caused
either by bright surface or by background a priori profiles. For such cases, systematic errors are smaller. Note that
satellite column averaging kernels, defined as the Box-AMF divided by the total AMF (Eskes and Boersma, 2003),
are provided for every observation. They can be used to remove the smoothing error component when comparing
the satellite data to any other external data.






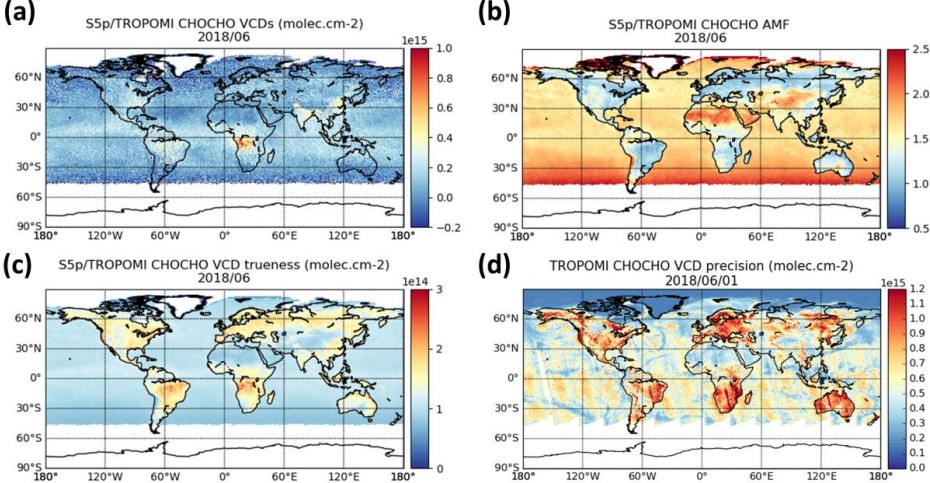


**Figure 5: (a) TROPOMI June 2018 monthly means of glyoxal tropospheric columns, glyoxal air mass factors (panel (b)) and glyoxal tropospheric column systematic errors (panel (c)). Scenes contaminated by clouds or Ice/snow have been filtered out. Panel (d) shows glyoxal tropospheric column random errors for one single day, in which all observations have been kept to illustrate the impact of the scene brightness.**

## 4. Comparison with other satellite instruments

### 4.1. Algorithmic differences for GOME-2A/B and OMI glyoxal retrievals

Glyoxal tropospheric columns have also been retrieved from other satellite instruments, namely GOME-2 on board the platforms Metop-A and –B and OMI on board AURA. Retrieval settings very similar to those described in the previous section were applied. For GOME-2A and B, we use data records recently produced within the operational environment of the EUMETSAT AC SAF (Valks et al., 2020) . We list here the remaining differences with respect to the TROPOMI algorithmic baseline and the specificities for each instrument.

All data sets essentially share the same DOAS fit settings (reference cross-sections, fit window, polynomial degree…). The heterogeneity cross-sections are omitted for the GOME-2 and OMI retrievals. While the instrumental design of GOME-2 makes it weakly sensitive to scene heterogeneity, it would be beneficial for OMI to include similar cross-sections but that would imply a reprocessing of the complete slant column data set data with limited added-value for the large-scale comparison with TROPOMI that we present in the next subsection. For the GOME-2 instruments, we also fit two additional cross-sections representative of the instrumental sensitivity to light polarization as provided from the level-1 key data (EUMETSAT, 2011) as well as one pseudo cross-section to account for an along-track spectral resolution change occurring due to instrumental temperature change (Azam and Richter, 2015). Note that for GOME-2 the cross-sections are convolved with an instrumental slit function optimized as part of the wavelength calibration for every measured irradiance (De Smedt et al., 2015), which allows accounting for the known long-term drift of the GOME-2 instrument spectral response function.



Differences in air mass factor calculations consist only in using, over land, a priori profiles provided by
IMAGESv2, the chemical transport model predecessor of MAGRITTE, at the coarser resolution of 2.0°x2.5°. For
the GOME-2 instruments, we use the directionally dependent Lambertian-equivalent reflectivity database
produced by Tilstra et al. (2021) instead of the OMI database.
A background correction procedure is applied consistently with the one used for TROPOMI. The GOME-2
instruments being whiskbroom scanners, there is no destriping procedure as such but instead a viewing zenith
angle-dependent correction is applied, also relying on the slant columns retrieved in the Equatorial Pacific sector.
This correction may account for example for remaining biases related to the instrumental polarization sensitivity.
For both OMI and GOME-2, the row/VZA dependence does not show any obvious change along the orbit and the
corresponding correction thus relies only on the low latitude measurements.
Note that the OMI and GOME-2 glyoxal products are filtered for cloudy scenes using cloud fraction lower than
20% as taken from the $O_2$-$O_2$ (Veefkind et al., 2016) and OCRA (Lutz et al., 2016) cloud products, respectively.
The empirical correction for strong $NO_2$ absorption signature described in section 3.1 has been applied to those
instruments as well. In the following section, we compare the TROPOMI glyoxal retrievals with the OMI and
GOME-2A/B data sets. The OMI record covers the period 2005-2018, while GOME-2A/B span the periods 2007-
2017 and 2013-2020, respectively. OMI and GOME-2A records were interrupted when their respective quality
was degraded too severely and other instruments were available to continue the morning and afternoon time series.
**4.2.   Glyoxal satellite inter-comparison**
**4.2.1. Comparison of the noise level**
As mentioned before, the level of noise in the satellite glyoxal tropospheric column products is large compared to
the real signal. This is illustrated in the upper panel of Figure 6 which shows all individual clear-sky TROPOMI
glyoxal columns retrieved in the Pacific Ocean on June, 1st 2018 and plotted as a function of their latitude. The
scatter is significant ($\sigma \approx$ 5-7x10$^{14}$ molec/cm$^2$) with respect to the small glyoxal VCDs averaged in 5°-latitude
bins in this sector. The lower panel compares the standard deviation of the retrievals from TROPOMI, OMI,
GOME-2A and B in the same remote sector for the 1st of June of their respective first year of operation. The
scatter in the retrievals is directly linked to the instrumental signal-to-noise ratio, which is documented to be
around 500 for OMI (Schenkeveld et al., 2017), 1000 for GOME-2 (Zara et al., 2018) and 1500 for TROPOMI
(Kleipool et al., 2018). In practice, we see that the CHOCHO observation noise is indeed slightly larger for OMI,
that GOME-2B retrievals are noisier than those from GOME-2A, which have a level of noise similar to
TROPOMI. Considering the very small footprint size of TROPOMI (3x7.5 km² and 3x5.5 km² after August 2019)
compared to the other instruments (GOME-2: 80x40 km²; OMI: 13x24 km² at nadir), the TROPOMI observation
noise is remarkably low. More importantly, the much larger amount of TROPOMI data compared to OMI (~15x)
and GOME-2 (~100x) allows maintaining a better time or spatial resolution for a given target noise level. For
example, the random error associated to the daily glyoxal column averaged in an area defined by a circle with a
radius of 50 km will be less than 0.5x10$^{14}$ molec.cm$^{-2}$ for TROPOMI, while it will remain larger than 2.5x10$^{14}$
molec/cm$^2$ and 4.0x10$^{14}$ molec/cm$^2$ for OMI and GOME-2, respectively.
This is illustrated in Figure 7, which compares January monthly mean glyoxal VCD fields over Asia at the
resolution of 0.05° for TROPOMI and OMI (upper panels) and 0.25° for GOME-2A and OMI (lower panels) after
one year of their respective operation. At the resolution of 0.05°, the level of noise in the TROPOMI glyoxal map
is very low and many details can be distinguished in the glyoxal spatial distribution. In particular, hot spots of
glyoxal over many megacities are clearly identified (e.g. over Bangkok, New Delhi, Ho Chi Minh City,
Shenzhen…) but also over Cambodia where large fires occur every year from January to March. At this resolution
of 0.05°, the level of noise in the OMI map remains high and prevents distinguishing such details. At the coarser
spatial resolution of 0.25°, the reduction of the noise in the OMI and GOME-2 monthly glyoxal fields appears to
be sufficient to better distinguish the glyoxal spatial distribution but at the cost of a significant smoothing. In the
next section, we will intercompare the four satellite products at low temporal and spatial resolution in order to
minimize the impact of the noise and to identify possible systematic discrepancies.

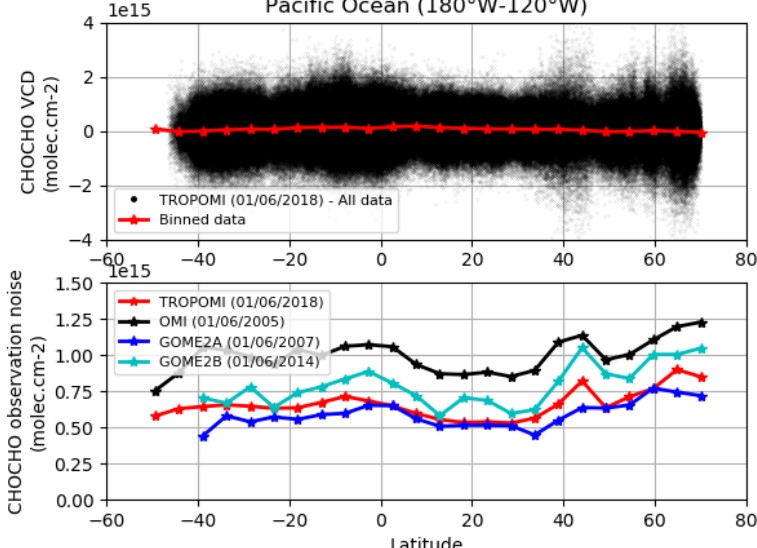


**Figure 6 : Illustration of the level of noise in satellite CHOCHO VCD retrievals. The upper panel shows all clear sky individual glyoxal VCDs retrieved in the Pacific Ocean from TROPOMI observations on June, 1st 2018. The scatter is very large compared to the low real signal as illustrated by the data binned in 5° latitude bands. The lower panel compares the standard deviation of the retrievals in the same sector from TROPOMI, OMI, and GOME2A/B on the 1st of June of their respective first year of operation.**

538



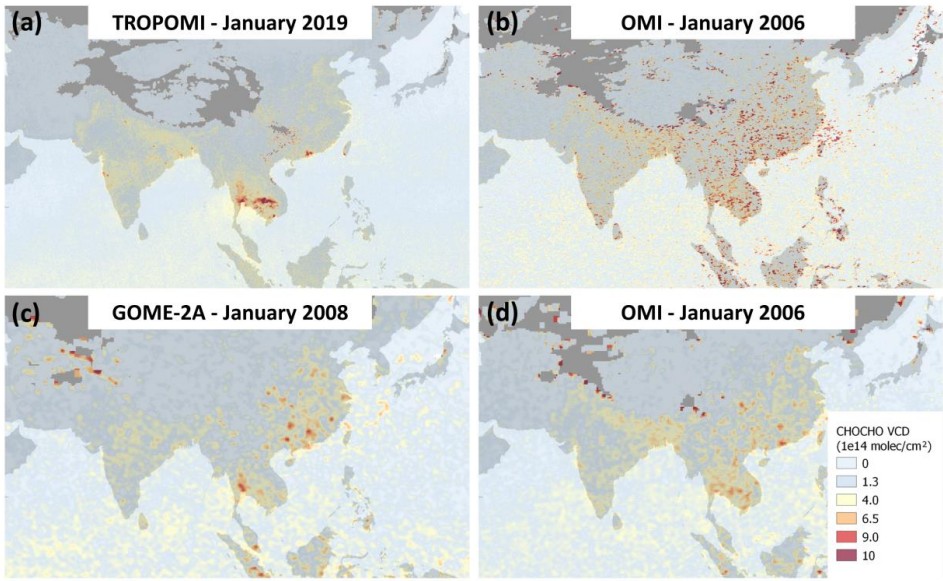

**Figure 7: Illustration of the impact of the instrumental signal-to-noise and available amount of data on monthly mean glyoxal VCD fields retrieved from different satellite instruments. The (a) TROPOMI data for January 2019 gridded at a resolution of 0.05°, (b) OMI data for January 2006 gridded at a resolution of 0.05°, (c) GOME-2A data for January 2008 gridded at a resolution of 0.25°, and (d) OMI data for January 2006 gridded at a resolution of 0.25°. Cloudy scenes have been filtered out and a smoothing filter has been applied on the four presented fields based on a spatial mean with the nearest neighbouring grid cells.**

### 4.2.2. Comparison of mean glyoxal fields

First, Figure 8 and Figure 9 compare seasonal maps of glyoxal VCDs generated from TROPOMI, OMI and GOME-2A/B data products. In order to reduce the data scatter for each instrument, those maps are based on long time series as indicated in the figures. Therefore, a one-to-one match is not expected. As can be seen, the consistency between the four instruments is excellent. Glyoxal patterns are captured similarly for all seasons in terms of both spatial distribution and VCD values. The largest glyoxal columns are observed in tropical regions, where biogenic emissions are important, and in regions with important fire events (e.g. Amazonia and Northern Africa in SON, Thailand/Indochina in MAM, Western US in August,...). At mid-latitudes, the glyoxal columns follow the seasonal cycle of biogenic activity with maximum values during summertime. Localized hot spots of glyoxal are visible over megacities corresponding to strong anthropogenic emissions (e.g. Northern China Plain, Bangkok, Teheran, New Delhi, Sao Paulo...). Also a persistent oceanic glyoxal signal is seen consistently by the four sensors. Note that a similar signal has been detected from ship- and airborne MAX-DOAS in the equatorial Pacific and Atlantic Oceans (Behrens et al., 2019; Sinreich et al., 2010; Volkamer et al., 2015). In contrast to TROPOMI and OMI, the level of noise in the GOME-2 data sets significantly increases over the South Atlantic Anomaly despite the application of a spike-removal procedure (section 3.1). Overall the GOME-2B maps are noisier than those from other sensors due to the lower signal-to-noise ratio of the spectra and a shorter time series. Compared to the UV, the sensitivity to the surface is larger in the visible, which may introduce interferences with





the spectral signature of specific ground surfaces, and thus may potentially lead to a bias on the retrieved columns.
A striking example is over the Kara-Bogaz-Gol near the Caspian sea, which is one of the saltiest lakes in the world
and contains large concentrations of sediments (Kosarev et al., 2009). The glyoxal signal detected over that lagoon
is unlikely to be physical and likely originates from interferences with the ground reflectance spectral signature.




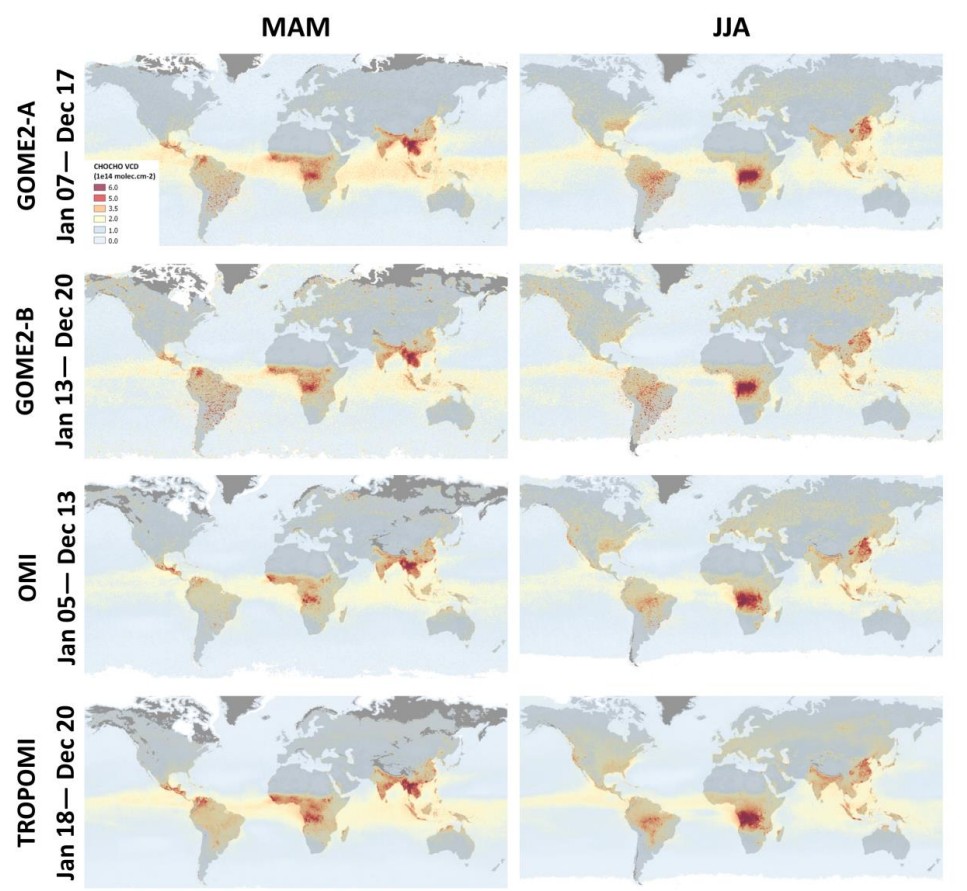


**Figure 8: Comparison of long-term averaged global CHOCHO VCDs (in $10^{14}$ molec/cm²) derived from**
**GOME-2A, GOME-2B, OMI and TROPOMI sensors, for the March-April-May period (left panels) and**
**the June-July-August period (right panels).**

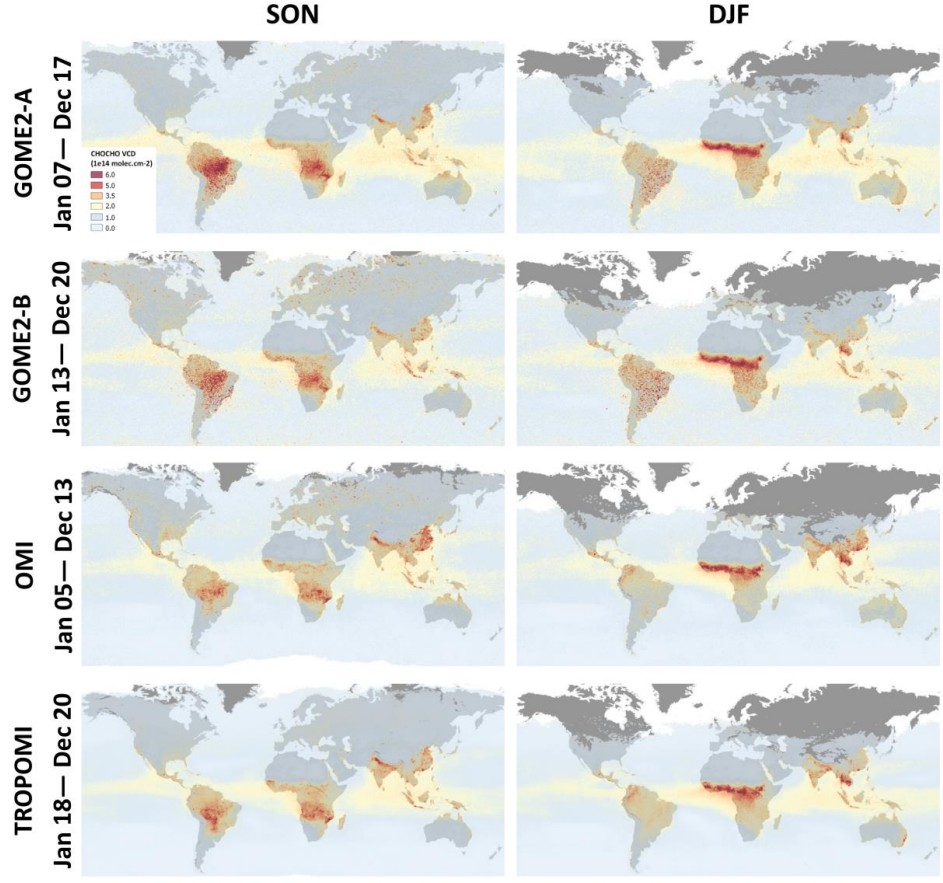


**Figure 9: Comparison of long-term averaged global CHOCHO VCDs (in $10^{14}$ molec/cm²) derived from GOME-2A, GOME-2B, OMI and TROPOMI sensors, for the September-October-November period (left panels) and the December-January-February period (right panels).**

For a more detailed investigation of the consistency of the TROPOMI data set with OMI and GOME-2A/B, we
compare complete time-series of monthly median glyoxal columns in selected regions (shown in Figure 10).
The red rectangles indicate the regions on which we focus in Figure 11 and Figure 12, while the global statistics
for all highlighted regions are given in Figure 13. Detailed figures are provided for all regions as supplementary
material (Figures S1, S2, S3, S4). Figure 11 compares directly the four full time series, while Figure 12
compares the typical climatological seasonal variations as obtained by combining all available years. The error
bars in the latter figure represent the interannual variability, and the 2-sigma standard deviation of the four
satellite products is indicated as inset.
In the Tropics (e.g. Amazonia, Equatorial/North Central Africa), the four data sets are relatively stable over time.
All instruments observe similar seasonal cycles and column values, although OMI appears to be slightly lower





than the others, in particular in Equatorial Africa. The inter-annual variability in Amazonia is high compared to
other regions worldwide. Glyoxal is produced in that region to a large extent by fire emissions, which are highly
variable. There is a direct correlation between years with high glyoxal columns and large fire emissions (e.g. 2007,
2010, 2015, 2019) as derived from the GFED database (van Der Werf et al., 2017;
https://www.globalfiredata.org/).Interestingly, glyoxal columns measured by the morning GOME-2 instruments
are larger than the OMI columns in the early afternoon during the fire seasons. This is consistent with the diurnal
variation measured in satellite HCHO columns by De Smedt et al. (2015) and would deserve further investigation.
Other regions display a more regular seasonal cycle, consistently seen by the four instruments.
In Asia, there are many hot spots, of which the origin is manifolds and strongly depends on the region and season.
In addition to biogenic activities, large emissions due to fires may significantly contribute to the glyoxal columns.
As illustrated in Figure 12, in the Indo-Gangetic Plain, there are typically two fire seasons in April/May and in
October/November (after the Monsoon period) related to agricultural burning of wheat residue (Kumar et al.,
2016), and leading to two maxima in the glyoxal VCD seasonal cycle with a significant interannual variability.
For example, during the COVID-19 Indian lockdown in April/May 2020, fire activity has been reduced leading
to smaller emissions (Levelt et al., 2021). This region is also highly populated, causing large emissions due to
human activities. This is also true in North-East China where glyoxal columns remain significant in winter, while
biogenic emissions are low during that period of the year. Although less variable than fire emissions,
anthropogenic emissions may also change over time. Despite those variable emissions, the four data sets spanning
different time periods show a high level of consistency. In China, it seems that the glyoxal columns as observed
by OMI, GOME-2A and B are slightly reduced after 2014. This would deserve further investigation. On the other
hand, any interpretation based on long-series of OMI data must be treated carefully since the instrument suffers
from an evolving row anomaly (Schenkeveld et al., 2017), which impacts the stability of the product and causes
an increasing number of outliers, especially at mid-latitudes. For example, over the Indo-Gangetic Plain, the OMI
columns deviate regularly from the other instruments after 2014. In general, remnants of noise are also visible in
the GOME-2 time series, which show somewhat less smooth time series than TROPOMI.
At mid-latitudes, the lower sun elevation, especially during local wintertime, makes the retrievals more
challenging. Nevertheless, a small maximum is consistently observed during the local summertime. During
wintertime, TROPOMI columns appear slightly lower than those from the other satellites. As mentioned before,
the stronger impact of the row anomaly at mid/high-latitudes leads to a larger number of outliers in the OMI data
set and to a low bias in winter after 2013/2014.




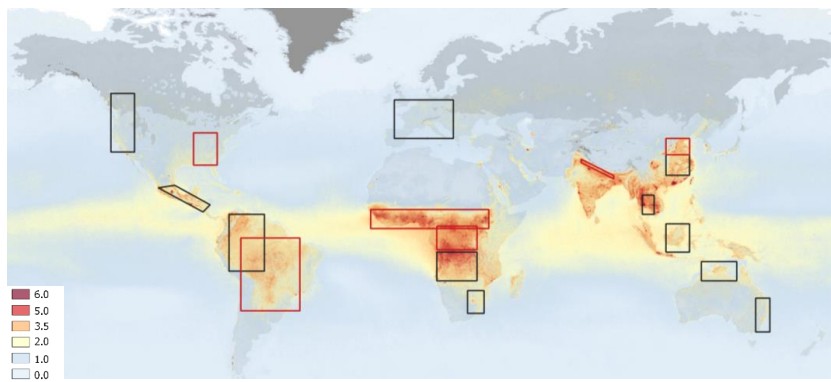

**Figure 10 : TROPOMI glyoxal VCD distribution (in 1e14 molec/cm²) averaged on the period January 2018-December 2020. The rectangles represent the regions where the glyoxal products from different satellites are intercompared with a specific focus on red regions in Figure 11 and Figure 12.**

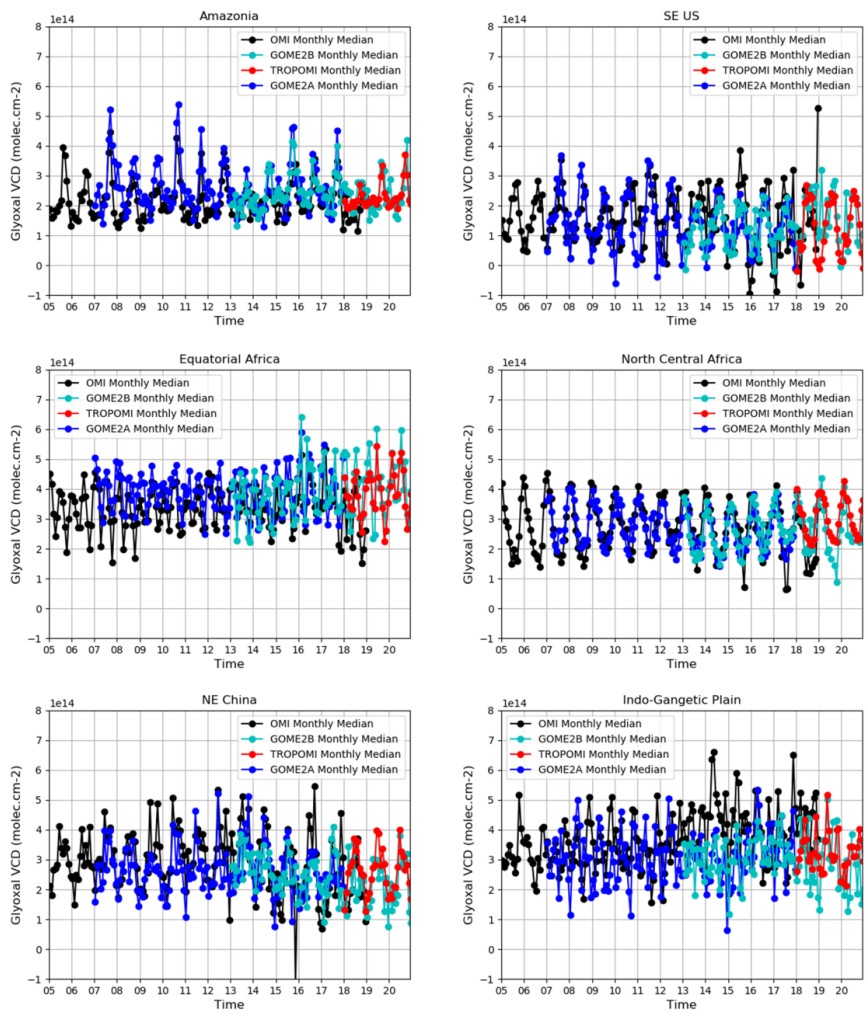

626

627    **Figure 11: Comparison of the monthly median glyoxal VCD time series from GOME-2A/B, OMI and TROPOMI in a**

628    **few selected regions worldwide.**

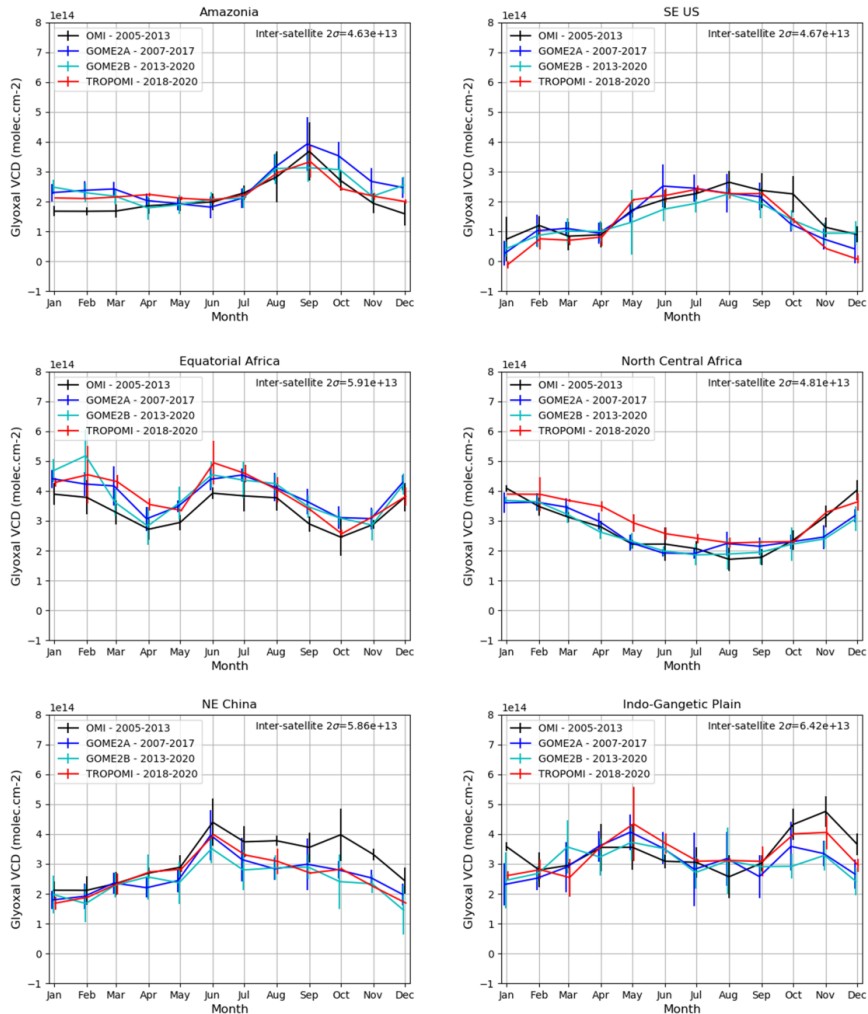

**Figure 12: Comparison of the climatological seasonal variation of the monthly median glyoxal VCDs from GOME-2A/B, OMI and TROPOMI in a few selected regions worldwide. The error bars represent the interannual variability as derived from the full time series.**

Figure 13 summarizes for all regions drawn in Figure 10 the absolute and relative deviation of each of the four data sets with respect to the median values of the ensemble. The symbols represent the median deviation considering all months of the year, while the error bars represent the full range of the monthly deviations. Regions are sorted by increasing mean glyoxal vertical column amounts and light and dark blue shaded areas indicate $2.5 \times 10^{14}$ molec/cm² (50%) and $1.5 \times 10^{14}$ molec/cm² (30%) differences as guidelines. Inter-satellite deviations are generally less than $5 \times 10^{13}$ molec/cm² (20%). The large error bars in the relative differences plot for mid-latitude regions are caused by local wintertime months during which the glyoxal content is very low, if not negligible, and are therefore meaningless. Overall, the inter-satellite consistency of the glyoxal VCD products is excellent. In the next section,





641 we will investigate the product quality with comparisons with independent ground-based MAX-DOAS glyoxal

642 observations at a few stations in Asia and Europe.

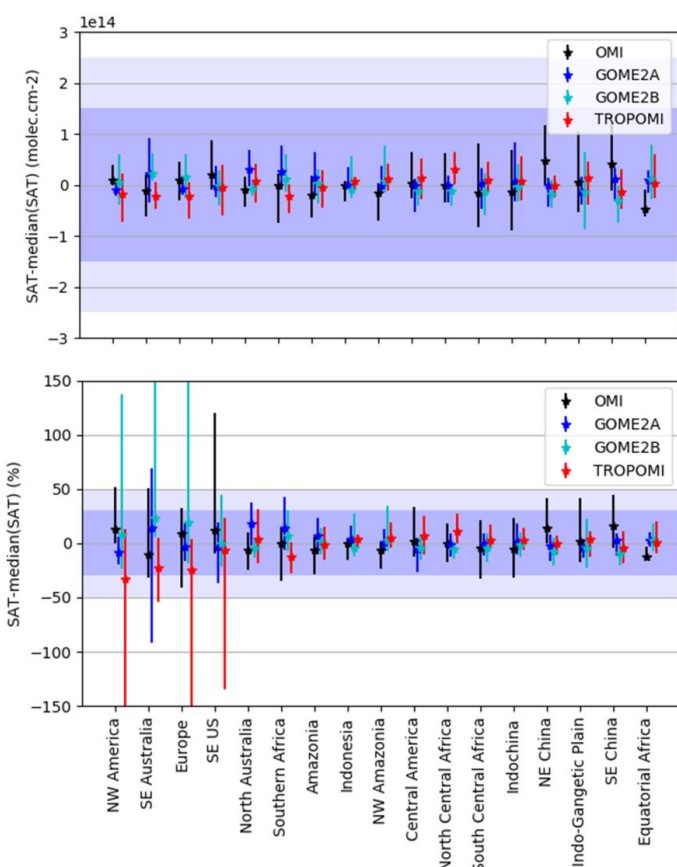

643

**Figure 13: Median deviation of the glyoxal VCD differences for TROPOMI, OMI, GOME-2A/B against the median value of the ensemble of the four data sets in the selected regions worldwide drawn in Figure 10. Those are plotted in absolute values (molec/cm²) in the upper panel and in relative values (%) in the lower panel. The error bars indicate the full range of the deviations considering climatological monthly data. Regions are sorted by increasing median glyoxal VCD value from left to right. The light and dark blue shaded area indicate differences of 1.5 molec/cm² (30%) and 2.5 molec/cm² (50%).**


## 5.   Validation with MAX-DOAS data

### 5.1.   Description of MAX-DOAS data sets and methodology


MAX-DOAS instruments measure scattered solar light in the UV-Visible spectral range at different elevation
angles above the horizon and allow retrieving information on trace gases and aerosol extinction in the altitude





range below 2-3km of the atmosphere, where the instrumental sensitivity is the highest. In a first approximation,
vertical columns of boundary layer gases can be estimated from MAX-DOAS measurements using a simple
geometrical approach (Brinksma et al., 2008; Hönninger et al., 2004). More elaborated approaches exploit a set
of different elevation angles to derive information on the vertical distribution of the gas concentration with up to
4 degrees of freedom, resulting in more accurate vertical columns in the 0-4 km altitude range (e.g. Beirle et al.,
2019; Clémer et al., 2010; Irie et al., 2011; Friedrich et al., 2019).
Glyoxal concentrations can be derived from MAX-DOAS measurements in the visible range. However, the
number of glyoxal MAX-DOAS data sets is very limited, especially those covering a period long enough to allow
the validation of satellite data during entire seasonal cycles. Moreover, MAX-DOAS retrievals are affected by
similar difficulties as satellite retrievals (noise, spectral interferences). Here, we collected an ensemble of data
sets from nine stations located in Asia and Europe (see Table 3) spanning at least one year. Altogether a wide
range of glyoxal columns and emission regimes are covered by those stations. Unfortunately, the approach to
retrieve glyoxal from MAX-DOAS has not been homogenized so far, and they cannot be considered as true
fiducial reference measurements. For example, although the same interfering species have been included in the
DOAS fits, the reference cross-section data as well as the fitting interval may vary. The design (spectral range,
spectral resolution, detector type, etc.) and operation mode of the instruments differ substantially, resulting in
different sensitivities to changes in retrieval settings. Finally, the slant-to-vertical column conversion is performed
differently from one station to another (see Table 3). Despite those limitations, the comparison of glyoxal
tropospheric columns from satellites with nine different MAX-DOAS instruments is unprecedented.
Among the available MAX-DOAS data sets, three (Xianghe/China, Chiba/Japan and Phimai/Thailand) are long
enough to allow a comparison with OMI and GOME-2A/B in addition to TROPOMI. The other ones span shorter,
and more recent periods and will be used only for comparison with the TROPOMI product. The Xianghe station
has the longest and stable data record, and provide vertical profiles of glyoxal. Therefore we have used this
reference station to perform a thorough analysis of the satellite product stability and of the impact of applying
satellite averaging kernels. For the other stations, we performed a more qualitative comparison of the seasonal
cycles of the glyoxal tropospheric columns. For the data colocation, we select MAX-DOAS data ±1.5 hour around
the satellite overpass time and satellite data within a radius of 100 km (150 km for Phimai) and 20 km around the
station for GOME-2A/B/OMI and TROPOMI, respectively. Daily median glyoxal columns are computed if both
satellite and ground-based data are available and finally monthly medians of the daily median columns are
compared.
**Table 3 : List of MAX-DOAS stations used in the study and brief description of the approach to generate the glyoxal**
**data.**

| Station (coordinates) Time range | Institution PI | Retrieval Approach and fit interval | Reference |
|---|---|---|---|
| Xianghe/China (39.75°, 116.96°E) 2010-2020 Uccle/Belgium (50.78°N, 4.35°E) 2017-2020 | BIRA-IASB | Profile retrieved using an Optimal Estimation scheme<br><br>436-468 nm | (Clémer et al., 2010; Hendrick et al., 2014) |
| Chiba/Japan (35.63°N, 140.10°E) 2012-2020 | CERES | Profile retrieved using a parametrization approach | (Hoque et al., 2018; Irie et al., 2011) |



| Phimai/Thailand (15.18°N, 140.10°E) 2014-2020 Pantnagar/India (29.03°N, 79.47°E) 2017-2020 | | 436–457 nm | |
|---|---|---|---|
| Mohali/India (30.67°N, 76.73°E) May 2019 - 2020 | MPIC/IISERM | Profile retrieved using a parametrization approach 400-460 nm | (Beirle et al., 2019; Kumar et al., 2020) |
| Athens/Greece (38.05°N, 23.80°E) 2018-2020 Vienna/Austria (48.18°N, 16.39°E) 2018-2020 Bremen/Germany (53.11°N, 8.86°E) 2018-2020 | IUP-UB | Columns retrieved using the Geometrical Approximation 436-468 nm | (Alvarado et al., 2020b; Gratsea et al., 2016; Schreier et al., 2020) |



## 5.2. Validation results


Figure 14 focuses on the comparison of monthly median glyoxal tropospheric columns retrieved from TROPOMI,
OMI, GOME-2A and GOME-2B with columns from the BIRA-IASB MAX-DOAS instrument in Xianghe
(China). The left panels compare the full time series for each satellite sensor with the MAX-DOAS data record.
The right panels show the corresponding satellite/MAX-DOAS absolute differences. Note that the MAX-DOAS
measurements have been interrupted from mid-2018 to mid-2019 due to an instrumental problem. Overall, all four
satellite instruments reproduce quite well the seasonal cycle seen by the MAX-DOAS instrument. However for
all of them, except for the recent TROPOMI, a degradation appears after a few years of operation. For OMI, the
consistency with the MAX-DOAS is excellent before 2013, but the number of outliers increases afterwards and
the columns during wintertime become too low. This is attributed to the evolving row anomaly as discussed in
section 4.2. The GOME-2A/B data sets also agree quite well with the ground-based data for their first years of
operation but then suffer from an increasing number of outliers after 2014 and 2017, respectively. Nonetheless,
the quality of the data sets remains very reasonable. The consistency of the TROPOMI time series with the MAX-
DOAS is also excellent and is characterized by a smooth temporal variability without any outliers on a monthly
basis. The absolute differences shown in the right panels also clearly indicates a reduced scatter compared to the
other satellites, despite the fact that a smaller overpass radius of 20 km was used instead of 100 km. This is
reflected in the standard deviation of the differences given in the titles of each subpanels. The TROPOMI standard
deviation is $0.9 \times 10^{14}$ molec/cm², while it is larger than $1.7 \times 10^{14}$ molec/cm² for other sensors. On average, there
are small negative biases with respect to the MAX-DOAS data for the four satellite time series (also given in the
panel titles), ranging between $-0.8 \times 10^{14}$ molec/cm² for TROPOMI and $-1.5 \times 10^{14}$ molec/cm² for OMI. For this
particular station, we investigated the impact of applying the satellite averaging kernels to smooth the MAX-
DOAS glyoxal profiles. This process allows simulating MAX-DOAS columns which would be retrieved from the
satellite algorithm, considering its own a priori profile information. The comparison of the satellite columns with
the smoothed MAX-DOAS data therefore removes differences due to imperfect satellite a priori profile
information. As shown in Figure 14, smoothing the MAX-DOAS columns reduces the satellite/MAX-DOAS bias
to values ranging from -0.2x10$^{14}$ molec/cm² (TROPOMI) to -0.5 x10$^{14}$ molec/cm² (OMI).

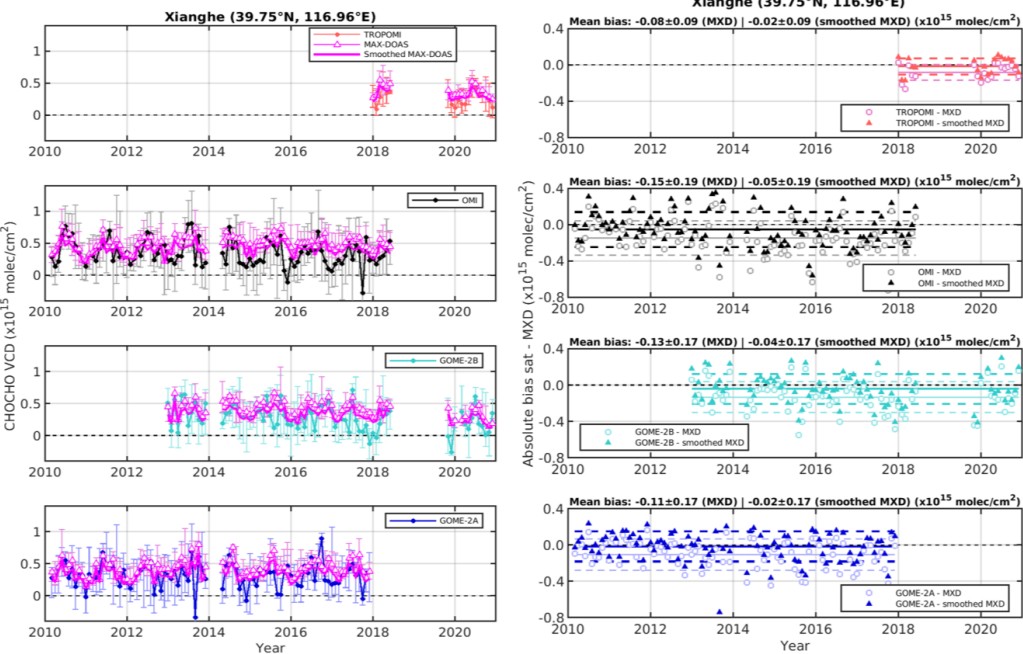


**Figure 14 : Comparison of the monthly median glyoxal tropospheric vertical columns retrieved from satellite and
MAX-DOAS (MXD) instruments in Xianghe (China). The four left panels compare the time series from TROPOMI,
OMI and GOME-2A/B with the MXD time series. MXD columns are also shown when smoothed with the satellite
averaging kernels. The error bars represent the 25 and 75% percentiles. The four right panels show the corresponding
time series of the satellite-MD absolute differences. Both original and smoothed MXD data are shown. Mean bias and
standard deviation of the differences are given in the panel titles and are also represented in the right panels with the
full and dashed coloured lines.**


In Figure 15, we compare the median satellite and MAX-DOAS seasonal cycles of the glyoxal tropospheric
columns at three stations (Xianghe, Chiba and Phimai) where the time series present a good overlap with the OMI
and GOME-2A and B records, in addition to TROPOMI. In Xianghe, the seasonal cycle of the smoothed MAX-
DOAS columns is also shown, illustrating again the reduction of the satellite/MAX-DOAS bias when the a priori
profile error component is removed. Note that the OMI and GOME-2B seasonal cycles are computed using data
until end of 2013 and 2016 to limit the impact of the increasing number of outliers. In each comparison panel, the
MAX-DOAS cycle is always computed using the same time range as the satellite instrument. Overall, the seasonal
patterns are consistently captured by the satellite and MAX-DOAS instruments. In Xianghe, the GOME-2A and



TROPOMI cycles follow closely the MAX-DOAS curves, although TROPOMI slightly underestimates the MAX-
DOAS columns during winter months. OMI and GOME-2B also reproduce the general seasonal pattern but show
a somewhat more scattered curve, likely due to their slightly less stable time series. In Chiba where the glyoxal
signal is mostly driven by biogenic emissions, the agreement between the satellites and the MAX-DOAS
measurements is excellent both in terms of variability and absolute values. Again, OMI shows a larger scatter (as
also indicated by the larger error bars representing the inter-annual variability). In Phimai, where pyrogenic
emissions are responsible for large glyoxal columns especially in the first few months of the year, the seasonal
variability seen by the satellites and the MAX-DOAS is very consistent. A negative bias larger than for other
stations is nevertheless observed. This can be related to other studies that identified larger biases in $NO_2$ or HCHO
DOAS products for elevated column conditions (e.g. De Smedt et al., 2021; Verhoelst et al., 2021; Vigouroux et
al., 2020). Possible causes for such biases are the different air masses probed by the satellite and ground-
based instruments, their different vertical sensitivity as well as the a priori vertical profile information
used in the retrieval algorithms.

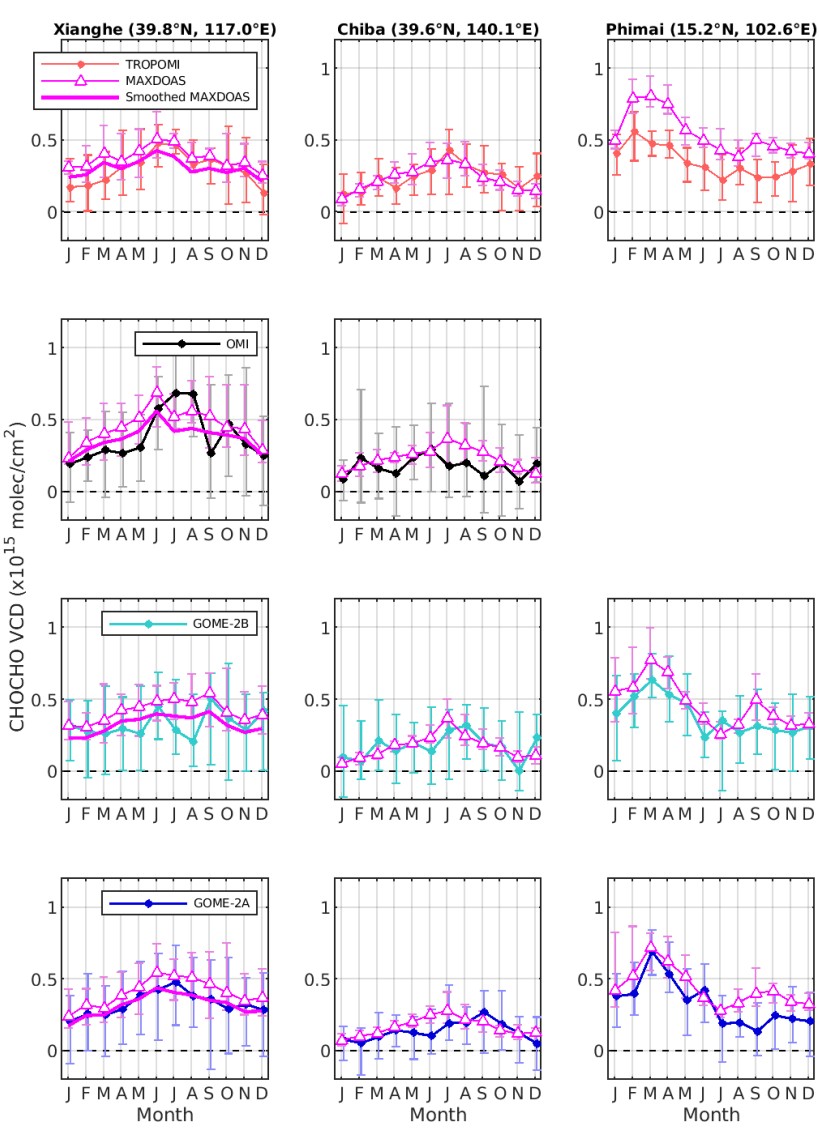

**Figure 15 : Comparison of the monthly median glyoxal tropospheric vertical column seasonal cycle as retrieved from TROPOMI, OMI, GOME-2A/B and MXD in Xianghe (China), Chiba (Japan) and Phimai (Thailand). The columns correspond to the three stations and the rows to the different satellites. In Xianghe, MXD data smoothed with the satellite averaging kernels are also shown. The error bars represent the interannual variability (25% and 75% percentiles based on the full time series available). Note that the comparison of with the MAX-DOAS data in Phimai is not shown as the latter starts in 2014 when OMI is degraded.**





In Figure 16, we compare again the seasonal cycle of glyoxal VCDs retrieved from TROPOMI with that from
more recent MAX-DOAS time series at six different stations. Four of them are located at mid-latitude in Europe
and show relatively low glyoxal columns, while larger average values are measured at the two other stations, in
Northern India (Mohali and Pantnagar). In Vienna/Austria and Athens/Greece, TROPOMI and MAX-DOAS
glyoxal columns agree very well and show consistent seasonal dependencies with maximum and minimum values
during summertime and wintertime, respectively.  On the other hand, at the higher latitude stations of
Bremen/Germany and Uccle/Belgium, the consistency of the seasonal variations seen from space and from the
ground is somewhat poorer. While the glyoxal columns agree well during summertime, the satellite columns tend
to underestimate MAX-DOAS values in winter, the latter showing almost no seasonal variation. Satellite glyoxal
retrievals at those latitudes are challenging in winter because of the low sun elevation causing a reduced sensitivity
to the lowermost atmospheric layers. As mentioned in section 3.2, observations with solar zenith angles larger
than 70° are filtered for this reason, which explains the gap between November and January at those two stations.
In Uccle, we have also tested the impact of smoothing the MAX-DOAS columns with the satellite averaging
kernels (similarly as for Xianghe), which turned out to be very small. The absence of any seasonal dependence in
the cities of Brussels (Uccle) and Bremen, in contrast to that observed (although limited) in Vienna and Athens,
is to some extent puzzling. One should keep in mind however that the glyoxal retrievals from MAX-DOAS
measurements are also challenging and it cannot be excluded that errors in ground-based data might also partly
contribute to the observed differences.
In Mohali and Pantnagar, glyoxal columns are much larger and the seasonal variability is driven by fire emissions
and meteorological factors such as the monsoon. At those two stations, the glyoxal seasonal variability is very
well reproduced by TROPOMI. In terms of absolute values, the TROPOMI columns agree reasonably well in
Mohali but, they significantly underestimate the (large) MAX-DOAS columns in Pantnagar. The reason why the
systematic satellite/ground-based bias is so different between those two stations is unclear. MAX-DOAS columns
are clearly higher in Pantnagar than in Mohali pointing either to possible local differences in air quality, not
reflected in the satellite data, or to inconsistencies in the ground-based data sets. Although the agreement is
excellent in Mohali, the typical behaviour is an underestimation of the columns by the satellites, as discussed
before. Note also that those sites are significantly contaminated by aerosols, which are neglected in the satellite
retrievals (apart from the stringent cloud filtering). MAX-DOAS data have also been analysed using very different
approaches, which may also cause differences. This calls for a more detailed analysis, which would require an
homogenization of the MAX-DOAS data treatment, a more sophisticated approach for the computation of the
satellite AMFs (e.g. with an explicit aerosol treatment) and possibly some independent information on the glyoxal
vertical distribution. This being said, the nice consistency in the glyoxal column seasonal variability by the
different systems is remarkable in itself. Table 4 provides an overview of the correlation coefficient between the
satellite and the MAX-DOAS glyoxal columns at all considered stations. For stations where the analysis was
possible for all satellite sensors, the correlation coefficient was found to be significantly better for TROPOMI
than for the other instruments. It is also clear that correlation coefficients are better for sites characterised by large
and highly variable glyoxal columns (e.g. Asian stations). Apart from the Bremen station where the negative bias
during winter leads to a low correlation coefficient, all other values are quite reasonable (between 0.61 and 0.87)
for TROPOMI. Table 4 also gives the mean bias as derived from the comparison of the satellite and MAX-DOAS
glyoxal column seasonal cycle as well as the standard deviation of the differences. As discussed above, the mean





differences are generally lower than $1 \times 10^{14}$ molec/cm², except for high columns where differences are noticeably
higher.

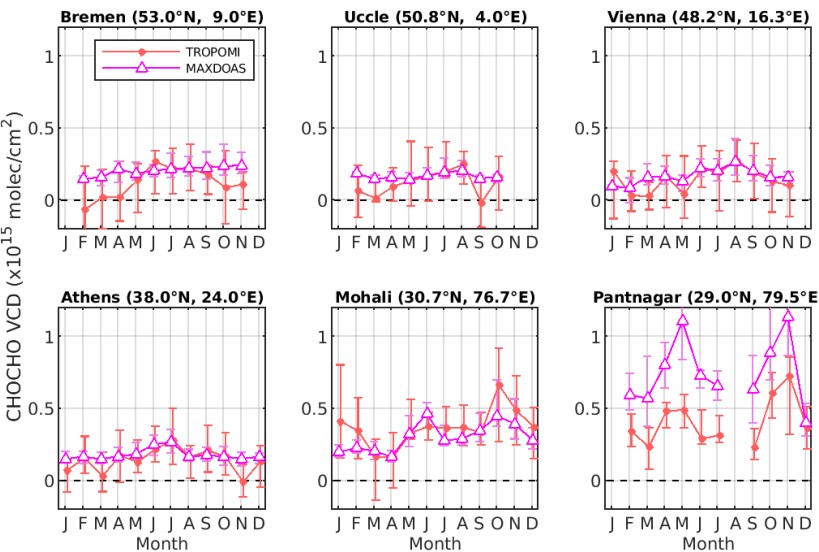


**Figure 16 : Comparison of the monthly median glyoxal tropospheric vertical column seasonal cycle as retrieved from**
**TROPOMI and MXD at four European stations (Bremen, Uccle, Vienna, Athens) and at two Indian stations (Mohali,**
**Pantnagar). The error bars represent the interannual variability (25% and 75% percentiles based on the full time**
**series available).**

**Table 4 : Correlation coefficients between the satellite and MAX-DOAS monthly median glyoxal tropospheric**
**vertical columns as well as mean absolute difference and associated standard deviation at nine stations.**

| | Correlation coefficient Mean bias ± standard deviation ($\times 10^{14}$ molec/cm²) | | | | | | | | |
|---|---|---|---|---|---|---|---|---|---|
| | **Xianghe** | **Chiba** | **Phimai** | **Bremen** | **Uccle** | **Vienna** | **Athens** | **Mohali** | **Pantnagar** |
| **TROPOMI** | 0.87 -0.8±0.6 | 0.80 0.1±0.6 | 0.85 -2.0±0.8 | 0.13 -0.9±0.9 | 0.67 -0.5±0.7 | 0.73 -0.3±0.6 | 0.61 -0.4±0.6 | 0.70 0.6±0.9 | 0.78 -3.5±1.5 |
| **OMI (until 2013)** | 0.70 -0.7±1.3 | 0.32 -0.6±0.8 | N/A | | | | | | |
| **GOME-2B (until 2016)** | 0.37 -0.9±0.9 | 0.66 0.0±0.7 | 0.88 -0.8±0.8 | | | | | | |
| **GOME-2A** | 0.92 -0.8±0.4 | 0.58 -0.1±0.9 | 0.86 -1.1±0.8 | | | | | | |



## 6. Conclusions

We presented the first global TROPOMI glyoxal tropospheric column product derived from three years (2018-2020) of visible radiance measurements. The DOAS-based algorithm, which relies largely on previous developments for heritage satellite nadir-viewing instruments, has been further improved in different aspects. In particular, the use of additional pseudo cross-sections in the DOAS spectral fit allows mitigating the effect of the instrumental spectral response function perturbations in case of scene brightness inhomogeneity, which otherwise would lead to systematic biases in the retrieved glyoxal columns. This helps removing artefacts along the coasts and reducing pseudo-noise in regions covered by persistent broken clouds. The glyoxal slant columns are also empirically corrected for biases caused by the $NO_2$ misfit in case of strong absorption. Finally, the background correction procedure has been optimized for the TROPOMI characteristics and the a priori glyoxal vertical distribution, essential to the AMF computation, is now provided by the CTM MAGRITTE, an updated version of the IMAGES model, running at the higher spatial resolution of $1°x1°$. The glyoxal column retrievals have been fully characterized with an error budget considering the different error components introduced in each of the algorithm modules. This allows extending the glyoxal column data product with total random and systematic error estimates provided for every observation, with corresponding averaging kernels and a priori profiles.

Glyoxal tropospheric columns have also been derived from data of the OMI, GOME-2A and GOME-2B satellite instruments using retrieval baselines similar to the TROPOMI algorithm. An extensive inter-comparison of those four data sets emphasised their excellent consistency with absolute mean glyoxal column differences found to be generally lower than $0.5x10^{14}$ molec/cm². This demonstrates that glyoxal retrievals respond in the same manner to our selection of settings for all nadir-viewing satellite instruments. Because of this sensitivity, the retrievals may be easily impacted by spectral features caused by instrumental degradation. We have shown that the stability of the OMI and GOME-2 data records is somewhat degraded after a few years of operations. Glyoxal retrievals are characterized by a high level-of-noise, requiring significant spatio-temporal averaging to extract meaningful signals. With both a much larger number of observations and a finer spatial resolution, TROPOMI outperforms by far the previous instruments in its ability to provide high quality and detailed glyoxal fields.

Satellite observations have also been compared with a few independent MAX-DOAS data sets from stations located in Asia and Europe. Owing to the scarcity of MAX-DOAS glyoxal data sets, especially covering several seasons, this validation exercise is therefore unprecedented. Based on a thorough analysis at the Xianghe station (China), where a 10-year time series of MAX-DOAS data is available, and on the comparison of seasonal cycles at other stations, we conclude that satellite and MAX-DOAS instruments observe consistent glyoxal signals and have similar intra-annual variations. This is reflected by the strong correlation coefficients, ranging between 0.61 and 0.87 for TROPOMI, with the exception of one mid-latitude station where the correlation is poorer. In general, the satellite and MAX-DOAS columns also agree in absolute values with differences less than $1x10^{14}$ molec/cm², at least for stations with moderate columns. In Xianghe, we showed that the application of the satellite averaging kernels to the MAX-DOAS data further reduces the mean differences. There are however two stations (Phimai/Thailand and Pantnagar/India) where the satellite/MAX-DOAS bias is more significant, despite an excellent agreement between the seasonal variations. The origin of this bias is not fully understood, but it is not uncommon to have such biases in UV-Visible satellite retrievals for strongly polluted sites. In addition, we have indications that the satellite observations are low-biased during wintertime at mid-high latitudes where both the



glyoxal signal is weak and the sensitivity to the boundary layer is reduced. The comparisons of OMI, GOME-2
and MAX-DOAS glyoxal columns also show reasonable agreement and similar intra-annual variability. Both the
correlation coefficients and the scatter of the satellite/ground differences were however less good than those of
TROPOMI. This points again to the better performance of TROPOMI for the detection of glyoxal from space and
to its enhanced capability at providing information on VOC emissions. For future work, it would be beneficial to
dedicate more efforts in the homogenization of the MAX-DOAS glyoxal retrievals in terms of both spectral
analysis and slant-to-vertical column conversion in order to strengthen their potential for the validation of satellite
data sets such as the one presented in this work.
**Data availability**
Access to TROPOMI glyoxal tropospheric column data is possible via the GLYRETRO website
(https://glyretro.aeronomie.be/), OMI glyoxal data can be obtained on request from the authors. Information to
download the GOME-2/Metop-A and GOME-2/Metop-B glyoxal data records is provided at
https://acsaf.org/datarecord_access.php.
**Author contributions**
CL is the main contributor to the study and led the writing of this paper. FH performed the validation exercise,
with support from MVR and LMAA. MVR, LMAA, AR, IDS, NT, JV, HY and JVG contributed to algorithm
and/or code development. TS and JFM provides the a priori modelled glyoxal profiles. PV and DL are responsible
for the production of the GOME-2 glyoxal operational data records. MVR, FH, LMAA, SFS, HI, VK, TW, VS,
TiW and PW contributed to operating the MAX-DOAS instruments, and to producing and providing glyoxal data.
CR supervised the study. All co-authors have been involved into the discussion of results and the writing of this
article.
**Competing interests**
The authors have the following competing interests: Thomas Wagner is chief-executive editor of AMT. Andreas
Richter is executive editor of AMT. Diego Loyola, Andreas Richter, Michel Van Roozendael and Thomas
Wagner act as associate editors for AMT.

**Acknowledgments**
This work contains modified Copernicus Sentinel-5 Precursor satellite data (2018-2020). It has been supported
by the European Space Agency via the preparation of the Level-2 Prototype Processor of the future Copernicus
Sentinel-5 satellite (contract #4000118463/16/NL/AI) and via the GLYRETRO project, part of the Sentinel-
5p+Innovation programme (contract #4000127610/19/I-NS). EUMETSAT, the Belgian Federal Science Policy
Office (BELSPO) and the German Aerospace Center (DLR) are acknowledged for their respective financial
support of the GOME-2 algorithmic developments through the AC SAF Continuous Development and Operations
Phase (CDOP-3) and the ProDEx B-ACSAF contribution to the ACSAF. The Vienna MAX-DOAS instrument is
part of the VINDOBONA project, which is funded by the Austrian Science Fund (FWF): I 2296-N29, the German
Science Foundation (DFG): Ri1800/6-1, and A1 Telekom Austria. IISER Mohali Atmospheric Chemistry Facility
is gratefully acknowledged for supporting the MAX-DOAS operations in Mohali, India. We thank Caroline Fayt
and Christian Herman from BIRA-IASB for maintaining the Uccle and Xianghe MAX-DOAS instruments.



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
