# Peer review of "Glyoxal tropospheric column retrievals from TROPOMI, multi-satellite intercomparison and ground-based validation"

_Atmospheric Measurement Techniques, 2021_

## Author Comment (AC1)

**Reply to review #1 of our manuscript number amt-2021-158 'Glyoxal tropospheric column retrievals from TROPOMI, multi-satellite intercomparison and ground-based validation.'**

We gratefully thank the reviewer for the careful reading of our manuscript and for the very constructive comments. Below, the reviewer's text is given in black while our replies and description on how the comments have been addressed in the manuscript are given in blue.

The manuscript titled '**Glyoxal tropospheric column retrievals from TROPOMI, multi-satellite intercomparison and ground-based validation**' presents a global tropospheric glyoxal product from the new TROPOspheric Monitoring Instrument (TROPOMI) along with a new retrieval algorithm that has been applied to the other satellite-based instruments for an intercomparison. These results are validated with some continental ground-based observations. The manuscript is well structured, albeit lengthy. Overall, it presents a step ahead in creating a high-resolution database for glyoxal, which is currently missing. While certain sections are well detailed and discussed, other parts are glossed over, and selective studies from the past have been used to suggest that the new product is accurate. This is especially of worry over the oceanic region where no validation is presented.

While I do not wish to be negative about the manuscript, which is worthy of publication in AMT after modifications, I hope that the authors can improve on the current draft according to the comments below.

Major comments:

Two major changes are suggested to make the paper easier to read and to highlight the capabilities and shortcomings of the updated retrieval algorithm.

1) A comparison of the different satellite products does not offer much to the current paper. All the satellite products are generated using essentially the same DOAS settings in the updated BIRA-IASB retrieval algorithm. The high level of consistency is not surprising, considering that the products are analyzed in almost the same way. The small differences that arise because of the physical detectors, footprints, etc., are not unexpected and hence the amount of discussion on this does not seem justified. It makes the paper lengthier than necessary and does not give extra useful information.

We respectfully disagree with this comment. It is true that, from a theoretical point of view, it is not surprising to have a good consistency between different satellites when common retrieval settings are applied. In practice, the low glyoxal optical depth makes it sensitive to any distortion of the measured spectra and having limited discontinuities is not necessarily as straightforward as one could think. For example, Alvarado et al. (2014) identified systematic differences when comparing different satellites, which needed further investigation. To our knowledge, this is the first study reporting satellite timeseries with this level of consistency In addition, the section also discusses the stability of the GOME-2/OMI data sets which may be impacted by the instrumental degradation. Therefore, we think it's worth insisting on this, and to demonstrate that those different data sets can be used combined together, which is very relevant for the creation of climate data records. In addition, this section gives the opportunity to discuss

the spatial and natural variability of the glyoxal column fields, which is useful for glyoxal non-experts.

2) One of the highlights, which needs to be discussed in more detail, is the high CHOCHO VCDs observed over the ocean. The new product shows high values over the oceans, for which the peak is about half as much as the continental peaks. As the authors have mentioned, this is not explicable by the current known chemistry and sources. Indeed, even in highly productive waters, glyoxal and methylglyoxal are significantly undersaturated, and hence a direct source is not likely (Zhu and Kieber, 2019). Eddy covariance based observations show that the ocean surface is a sink for glyoxal for most of the day (Coburn et al., 2014).

This elevated column over the tropical oceans was reported earlier for satellite observations (Lerot et al., 2010; Vrekoussis et al., 2010)  and other older papers using SCHIAMACHY. However, only one group has reported high CHOCHO over one single region in the pacific when using ground-based, or aircraft-based observations (Sinreich et al., 2010) – it has not been seen by others, even in the same region.

The largest collection of ship and land-based observations using data from nine campaigns all over the marine environment have shown that CHOCHO is mostly below the detection limit in the open ocean environment (Mahajan et al., 2014). A pattern of a significant increase in the tropics was not seen. A similar result was also seen by a more recent study by (Behrens et al., 2019) which showed that CHOCHO was mostly below the detection limit with just two days of values just above the detection limit – with the geographical distribution not the same as the new satellite product. Indeed, remote ocean observations from outside the tropical region also show similar glyoxal levels as the tropical regions (Lawson et al., 2015).

Considering this, it would be helpful to have a section about the potential interferences over the ocean:

- What was the effect of the liquid water absorption and vibration Raman infilling of Fraunhofer lines in spectral retrieval over different regions? Are oceanic regions more sensitive than over land?
- How sensitive is the retrieval over the oceans to the chosen background?
- Are there reasons why the retrieval shows significant seasonal changes over the land but not over the ocean?
- Considering the scale of the TropOMI pixels, large lakes could be used as testbeds to check the algorithm's sensitivity to water reflectance-related issues.

Some of these issues, especially related to the liquid water path, can play a big role in false positives over the tropical oceans. A study using OMI has detailed this in the past and should be referred to (Chan Miller et al., 2014) when discussing the retrieval over remote oceans. They were able to correct the elevated retrievals to large degree.

We agree that the oceanic pattern should be discussed a bit more thoroughly and we thank the reviewer for raising this point and for the different ideas and references. We also agree that the origin of this pattern is under debate and that (at least part of) it might originate from spectral interferences. We took the different comments into account for revising the manuscript. Below is first a reply to each of them and we describe after how this has been included in the text.

The fit of the liquid water signature is mostly important in remote clear oceans where the light penetration depth in ocean is the most important (see Lerot et al, 2010, Chan Miller et al., 2014, Peters et al., 2014; doi:10.5194/amt-7-4203-2014). VRS also produces a signal in those remote oceans (Vasilkov et al., 2002; https://doi.org/10.1029/2002GL014955). Peters et al. (2014) have shown that the fit of the liquid water cross-section combined with an intensity offset efficiently correct for those effects. When not corrected for, they lead to negative glyoxal columns. In our product, we see that the presence of systematically negative columns over those areas is largely limited. Over equatorial oceanic columns where enhanced glyoxal columns are observed, the liquid water/VRS signal is low. There is therefore no clear link between the liquid water path and the enhanced columns.

Uncertainties in water vapour absorption cross-section, on the other hand, might lead to spectral interferences with glyoxal and explain part of the observed signal. Sensitivity of glyoxal columns to the choice of the water vapour cross-section and of the temperature to generate it is non-negligible. When conducting sensitivity tests, we have indeed noticed that using a lower temperature similarly to what Chan Miller et al. (2014) did (i.e. 280 K) leads a reduction of the glyoxal columns for areas with high $H_2O$ content, including equatorial oceanic oceans. However, effective $H_2O$ temperatures (computed as the mean of the ECMWF T° profiles weighted by $H_2O$ concentrations) are typically larger than 290K) in area with large $H_2O$ concentrations. For this reason, we decided to continue working with the higher temperature of 293K, which appears closer to the physical conditions. These sensitivity tests nevertheless point to a need for improved water vapour absorption data.

The suggestion made by the reviewer (1st minor comment) is interesting. Introducing such a (5-10 nm) gap would remove a large part of the second glyoxal absorption band and undoubtedly would require an extension of the fitting window towards the UV to maintain stable retrievals and limit the associated noise increase. In our work, we decided to stick to our original window since it showed good performance in the past (and in the work from other teams). It also avoids the intense lines of the Ring signal below 435 nm, which may also bias the glyoxal fits. Provided this important debate, it will however worth in future to further investigate this suggestion.

The retrieval algorithm is relatively insensitive to the choice of the reference sector. The equatorial Pacific is used for both computing the mean radiance used as the reference in the DOAS fit and for the destriping step in the background correction. This ensures that any systematic row-dependent bias which would be introduced by the reference mean radiances would be efficiently removed with the destriping. In addition, the final offset correction in the background correction is based on a large range of latitudes in the Pacific covering regions with low glyoxal columns. This stabilizes in time the overall background of the columns and avoids that the possible variability in the equatorial oceanic regions impacts significantly the product. There is obviously an uncertainty associated to the chosen reference value for the background normalization as discussed and included in the error budget (section 3.4.3.). The latter is as significant as the errors due to AMF or DOAS fits. In addition, all the intermediate variables for computing the glyoxal tropospheric vertical column are given in the product (SCD, AMF, corrected-SCD). An advanced user is then in position to recompute glyoxal VCDs using a different reference background value.

Regarding the inner lakes, the TROPOMI spatial resolution allows indeed to see details unidentified with previous instruments. As discussed at the end of section 4.2.2, the

sensitivity to the surface is larger in the visible than in the UV, which makes the retrieval more sensitive to ground surface signatures. Inner lakes have sometimes specific water characteristics with an associated specific "surface signature", which cannot be easily considered in our retrievals. This may impact the retrieved glyoxal columns which are sometimes high or low biased over such inner lakes. We illustrated this in the text with the example of the Kara-Bogaz-Gol near the Caspian sea, one of the saltiest lake in the world, over which enhanced glyoxal columns are measured. Other lakes show a negative signal (e.g. the Van Lake in Turkey). Those lakes are therefore unique case studies but their specificities prevent to use them as test cases easily generalizable to the oceans.

Corrections in the manuscript:

1. We have added one section 4.2.3 "Glyoxal over equatorial oceans "as suggested by the reviewer to further discuss the oceanic glyoxal pattern seen in our data sets and the apparent inconsistency with field data:

[revised manuscript text omitted]

Minor comments:

1. P:5, L:157-159: Authors have used 435-460 nm wavelength window for the CHOCHO retrieval. Although several groups have used this window, it can be affected by the strong water vapor absorption in this region. Have authors considered introducing a gap in the analysis window for $H_2O$ or tried using a larger window (some studies recommend the 410-460 nm, 400-60 nm, etc.) to check the sensitivity of the retrievals?

   Please see response above.

2. The authors mention that 'There are however two stations (Phimai/Thailand and Pantnagar/India) where the satellite/MAX-DOAS bias is more significant, despite an excellent agreement between the seasonal variations. The origin of this bias is not fully understood, but it is not uncommon to have such biases in UV-Visible satellite retrievals for strongly polluted sites.' – this needs to be explored in more detail – the explanation about high pollution does not check out as the match is better at other polluted stations like Xianghe/China and Mohali/India.

   The satellite glyoxal columns at those stations are indeed in the same range as some other stations (e.g. Xianghe), but the columns as seen by the MAX-DOAS instrument are significantly larger, especially in Pantnagar but also in Phimai during the first part of the year. We have already discussed the possible causes for this bias in the text, including a possible contribution from the MAX-DOAS themselves (lines 797-802; 833-843). In the conclusions, we have slightly rephrased the sentence mentioned by the reviewer as:

   "*There are however two stations (Phimai/Thailand and Pantnagar/India) where the satellite/MAX-DOAS bias is more significant, despite a reasonable agreement of the measured seasonal variations. Although the origin of this bias is not fully*

*understood, the MAX-DOAS columns at those stations are very high and it is not uncommon to have such biases in UV-Visible satellite retrievals for strongly polluted sites. It cannot be excluded that part of the bias originates from the MAX-DOAS retrieval strategy at those sites."*

3. Line 1274: Reference title is incomplete.

   This has been corrected

4. The authors use 'excellent' at several places – this is very subjective and in most matches are not even 'great' – please reduce the use of superlatives throughout the manuscript.

   Agreed. We have softened a bit the wording.

5. The font size in most figures is too small to read without zooming.

   We have improved the visibility of Figs. 3, 11, 12, 14, 15, 16, S1, S2, S3, S4.

6. Figure 15: The legends can be moved to the empty panel.

   We think it is clearer if each row has its own legend to show clearly that they correspond to the different satellites. The legend in the upper row has however been modified to avoid crossing the axis.

7. The ground-based instruments use different DOAS retrieval settings and algorithms. For consistency and standardized validation, should they not be analyzed with the same settings and algorithms?

   Ideally, it would be indeed much better to have MAX-DOAS data retrieved in a homogeneous way. The MAX-DOAS data sets have been provided by different teams and homogenizing and reprocessing them would require a major effort, which is out of the scope of this work. However, we stated in different places in the manuscript that it would be beneficial to dedicate some resources to such an activity in future.

---

## Author Comment (AC2)

**Reply to review #2 of our manuscript number amt-2021-158 'Glyoxal tropospheric column retrievals from TROPOMI, multi-satellite intercomparison and ground-based validation.'**

We gratefully thank the reviewer for the careful reading of our manuscript and for the very constructive comments. Below, the reviewer's text is given in black while our replies and description on how the comments have been addressed in the manuscript are given in blue.

The manuscript "Glyoxal tropospheric column retrievals from TROPOMI, multi-satellite intercomparisons and ground-based validation" by Lerot et al., presents global glyoxal observations made by the TROPOspheric Monitoring Instrument (TROPOMI). The paper provides the description of the retrieval algorithm, an inter-comparison of glyoxal observations from TROPOMI and other low earth orbit (LEO) satellite retrievals, and validation leveraging a few available MAX-DOAS glyoxal observations. This new retrieval would enable new atmospheric chemistry studies given improved spatial resolution and retrieval noise levels in comparison with retrievals from prior LEO satellites. The paper well written and constitutes a good reference for future studies using TROPOMI glyoxal observations. Its publication its therefore more than justified.

There a few aspects of the retrieval description, the uncertainty calculation and the comparisons with other satellite could benefit from further descriptions and clarification. It would be great if the authors could address the following comments during the discussion before final publication of the manuscript in AMT.

My main concern regarding the different retrieval steps is the assumption of a constant $1\times10^{14}$ molecules/cm² vertical column over the Pacific Ocean as reference for the background correction. This value is based on observations from one group (Sinreich et al., 2010) using an observation methodology similar to the satellite retrieval (DOAS fit) that could be affected by similar biases. At the same time, this results differ from other ocean glyoxal observations (for example Mahajan et al., 2014) reporting smaller columns over the oceans. It would be interesting to provide further discussion about the effect of the background correction in the final reported columns. How much would differ the final columns have the author's decided to use reference columns from chemical transport models or other sources?

We agree with this comment and the uncertainty related to the reference value within the Pacific sector. Below is first our detailed reply and then we describe how the manuscript has been modified to clarify this aspect.

The large uncertainty related to the reference value is included in the total error budget (section 3.4.3). We took an error associated to this reference value $\sigma_{N_{v,0,ref}}$ of $5\times10^{13}$ molec/cm²; which is propagated through the retrieval. Similarly, the errors on the AMFs and slant columns from the reference measurements are also taken into account to compute the total background correction error. As illustrated in the Figure C1 below for one S5p orbit passing over Africa, the background correction error (red plain curve) is significant and contributes as much as the DOAS (blue curve) and AMF (green curve) error components to the total glyoxal VCD systematic error (black curve). The figure also shows that the choice of the reference value $N_{v,0,ref}$ contributes to about half of the total background correction error.
The choice of the value for the reference glyoxal vertical column indeed directly impacts the overall level of the product, with some small modulation directly related to the ratio of

the AMFs over Pacific and in other regions ($M_0/M$ in eq. (1)). This is illustrated in Figure C2, which shows the impact of using a lower Pacific reference value on the final glyoxal VCDs.

It has to be noted that all the intermediate variables for computing the glyoxal tropospheric vertical column are given in the product (SCD, AMF, corrected-SCD). An advanced user is then in position to recompute glyoxal VCDs using a different reference background value if needed.

[Figure]

*Figure C1 : Illustration of the zonal mean total systematic CHOCHO VCD error with its different components for one S5p orbit passing over Africa.*

[Figure]

*Figure C2 : Impact on the mean CHOCHO VCD for one S5p orbit of using a reference Pacific column of 0.5E14 instead of 1E14 molec/cm². The upper panel compares the corresponding lower columns (in red) with the original columns (in black). The absolute differences plotted in the middle panel are anti-correlated with the AMFs shown in the bottom panel.*

Corrections in the manuscript:
1. After the first paragraph of section 3.3 the following sentences have been added:
   *"There is nevertheless an uncertainty related to this reference value, which*

*impacts the overall level of the product. This error component is further discussed in section 3.4.3 and is taken into account to estimate the total glyoxal VCD error. As all intermediate variables (SCD, corrected-SCD, AMF) are provided in the product, a user could recompute glyoxal VCDs using a different reference Pacific value."*

2. At the end of section 3.4.3, we have added: "*Using a different reference value would directly impact the overall level of glyoxal VCDs worldwide, with some small modulations related to the ratio of the AMFs over Pacific and in other regions following Eq. (2).".*

3. In section 3.4.4, we have added a simplified version of Fig. C1 and the corresponding description in the text: "F*igure 7 shows the zonally averaged total systematic error along with its different components for one S5p orbit passing over Africa. In general, the three components contribute similarly to the total error for emission conditions. On contrary, the AMF error becomes smaller in background conditions while the two other terms dominate.*"

Also, to understand the effect of each retrieval step in the final VCDs around the globe it would be beneficial to add a figure showing global values of dSCDs, VCDs, and background corrected VCDs so it is easier to interpret the amount of information present in the final VCDs brought in by each retrieval step.

This is a good suggestion and we have added a figure (Figure 1) in the general overview of the algorithm presenting the main output for one day of TROPOMI data of the different algorithmic steps, which are further described afterwards. The colorbar is slightly different than for other figures to better show possible structures in other ranges of values (see minor comment below).

**Other comments and doubts:**

The description about the calculation of pseudo-absorbers to account for scene heterogeneity leaves some questions un-answered: (1) what are the criteria defining the two additional cross-sections for scene heterogeneity? (2) What is the effect of using one vs. two extra pseudo cross sections? (3) how is defined the remote region over which the heterogeneity cross sections are calculated?

The heterogeneity factor is computed for every nominal ground pixel using the radiances available in the L1 data at higher spatial resolution for a limited number of wavelengths. This factor represents the scatter of the small pixel radiances within the nominal ground pixel. It generally ranges from -1 to +1 with values close to 0 indicating an homogeneous scene while higher absolute values indicate a larger level of brightness heterogeneity within the ground pixel. The heterogeneity cross-sections are constructed by comparing systematic residuals of scenes with a heterogeneity factor larger than +/- 0.08 with those from homogeneous scenes. Indeed the two cross-sections are significantly correlated. However, some remaining differences between them (likely related to the way the ISRF is perturbed depending on the radiance distribution within the ground pixel) lead to a further improvement of the fit quality when including the two.
The cross-section construction can be done using one single orbit over the Pacific. Within this process, we assume that the glyoxal fields vary smoothly along the orbit track.

In the manuscript, we have slight rephrased the section 3.1.1 to add those different pieces of information. It reads now as:

"*Those cross-sections are generated with a statistical analysis of the fit residuals for many observations in the Pacific Ocean as a function of the level of scene heterogeneity. The latter can be computed using radiance measurements at higher spatial resolution available in the TROPOMI level-1 data at a limited number of wavelengths. It ranges between -1 and +1 and is close to/deviates from 0 for homogenous/heterogeneous scenes, the sign indicating the part of the ground pixel that dominates the scene brightness. Following this approach, two additional cross-sections corresponding to the systematic residuals of scenes with an heterogeneity factor larger/smaller than +/- 0.08 have been added to the DOAS baseline and both the fit residuals and the identified glyoxal biases have been reduced as illustrated in the right panels (b) and (d) of Figure 2 . This effect is particularly visible along coasts and mountains but also over lands where some pseudo-noise caused by persistent broken clouds is also largely reduced. Although significantly correlated, including the two heterogeneity cross-sections leads to a further improvement of the fit quality, likely due to a slit function perturbation that depends on the radiance distribution within the nominal ground pixel.*"

How many Taylor expansion terms are considered in the derivation of the empirical correction associated with $NO_2$ slant columns?

We used a first order Taylor expansion of the NO2 optical depth around the wavelength and the vertical optical depth. Therefore, two additional cross-sections are used to derive the empirical correction.
The information has been added to section 3.1.2

Are the MAGRITTE a priori glyoxal vertical profiles computed daily at the satellite over pass time or are they compiled as a monthly climatology as done in most heritage glyoxal satellite retrievals?

Indeed, the a priori profiles are still based on climatological data. However, the model has been recently updated for the chemical and deposition mechanisms. In addition, the a priori profiles are provided by the model at the respective satellite overpass time. The information has been added in the text.

How is the interpolation of the background correction matrix done outside the 40°S to 40°N area?

To avoid meaningless correction values out of the 40°S-40°N area, no extrapolation is performed in the second step of the background correction but instead we simply use the nearest neighbour correction values. The final offset correction can be safely applied to all regions, even if only prescribed by the 40°S-40°N region. The information has been added in the text.

The classification of all AMF uncertainties as systematic is confusing. First, it is important to acknowledge how complicated it can be discriminate systematic and random uncertainties in the AMF calculation and the different sources of uncertainty. The authors should be thank for the efforts they have put in trying to quantify such

uncertainties. Said that, given the uncertainties inherent to chemical transport models and surface reflectance climatology, and the representation errors associated with different spatial and temporal resolutions some of the AMF errors have to be necessarily random. Given the mean biases between MAX-DOAS observations and TROPOMI retrievals reported in the manuscript (always $< 0.6 \times 10^{14}$ molecules/cm$^2$) should not the systematic uncertainties reflect this in panel c) of figure 5 with negative values?

We agree that this classification can significantly depend on the application and on the spatial and time resolution of interest.
Here, we considered as systematic the errors that would remain the same for a measurement at the exact same location and time, as well as same atmospheric conditions. As the reviewer says, when looking at extended temporal or spatial scales, part of those systematic errors may appear like noise and our approach to estimate the errors is therefore conservative. This has been discussed by Vigouroux et al. (2020) (https://doi.org/10.5194/amt-13-3751-2020), who attributed part of the scatter in HCHO satellite-MAXDOAS differences to a random component of the AMF errors.

The total systematic error is obtained by assuming that the different error components are uncorrelated and is derived as the root square of the quadratic sum of the different error components (eq (2)). Therefore, this estimate gives an indication of the possible range of the total systematic error without any indication of its sign. From this perspective, validation clearly provides crucial complementary information even if the identified absolute biases are consistent with the estimated errors.

We have added in the manuscript at the end of the 1st paragraph of section 3.4 the following statements:
"*It has however to be noted that the latter assumption may lead to conservative systematic error estimates and to an underestimation of the product scatter, depending on the time and spatial resolution of interest. In particular, uncertainties associated to the input parameters needed for the AMF calculation are directly related to the resolution of the used databases and may appear as random at coarser resolution. This has been discussed by Vigouroux et al. (2020) who attributed part of the scatter in formaldehyde vertical column TROPOMI/MAX-DOAS differences to a random component of the AMF errors.*"

During the discussion of uncertainties associated to a priori glyoxal profiles, an effective height uncertainty of 50 hPa is assumed. How is this value obtained?

The uncertainty of the effective profile height is determined by statistical analysis of the profile heights from one year of model data. 50hPa corresponds roughly to the standard deviation of the profile heights over polluted regions. The information has been added.

While the color scale used in figures 7, 8, 9, and 10 produce clean plots they fail to convey complete quantitative information. First, despite glyoxal VCDs ranging between 0 and $1 \times 10^{14}$ molecules/cm$^2$ in most parts of the world the color scheme does not allow appreciating any structure for that given range. Second, what color is assigned for values below 0 and above $6 \times 10^{14}$ molecules/cm$^2$?

Values smaller than 0 or larger than $6 \times 10^{14}$ molecules/cm² are colored with the colorbar extremes (light blue or dark red). Indeed, the colorbar of Figs 7-10 has been chosen to highlight at best the more important emission regimes. Nevertheless, we think that the

main persistent weak CHOCHO signals are also visible. In the new Figure 1 that we added to illustrate the different algorithmic steps, the colorbar is slightly different and for example better shows that the amount of negative columns over oceans is limited.

**Minor typos and language comments**:

Line 56: I think "precursors is" should be "precursors are"

This has been corrected.

Line 207: "end hence" should most likely by "and hence"

This has been corrected.

Line 316: "Anthropogenic NMVOCs emissions of are" should be "emissions are" without the "of"

This has been corrected.

Line 438: "see above section 6.5.1" is meaning "see section 3.3"?

We intended to refer to section 3.4.1. This has been corrected.

---

## Author Response (AR2)

**Changes made to account for final comments on the revised manuscript number amt-2021-158 'Glyoxal tropospheric column retrievals from TROPOMI, multi-satellite intercomparison and ground-based validation.'**

As requested, we have added a sentence in the abstract to highlight the current uncertainties associated to the glyoxal levels over equatorial oceans.

We replaced the sentence

"Systematic errors are typically in the range of $1\text{-}3\times10^{14}$ molec/cm² (~30-70% in emission regimes)."

by

"Systematic errors are typically in the range of $1\text{-}3\times10^{14}$ molec/cm² (~30-70% in emission regimes) and originate mostly from a priori data uncertainties as well as spectral interferences with other absorbing species. The latter may be at the origin, at least partly, of an enhanced glyoxal signal over Equatorial Oceans and further investigation is needed to mitigate them."